# Physically Aligned Hierarchical Mesh-based Network for Dynamic System Simulation

## Abstract

Dynamic systems evolve through complex interactions, where local events influence global behaviors, reflecting the interconnected nature of real-world phenomena. Simulating such systems demands models that effectively capture both local and long-range dynamics, while maintaining a balance between accuracy and computational efficiency. However, existing mesh-based Graph Neural Network (GNN) methods often struggle to achieve both high accuracy and efficiency, especially when dealing with large datasets, complex mesh structures, and extensive long-range effects. Inspired by how real-world dynamic systems operate, we present the Mesh-based Multi-Segment Graph Network (MMSGN), a novel framework designed to address these challenges by leveraging a physically aligned hierarchical information exchange mechanism. MMSGN combines micro-level local interactions with macro-level global exchanges, aligning the hierarchical mesh structure with the system's physical properties to seamlessly capture both local and global dynamics. This approach enables precise modeling of complex behaviors while maintaining computational efficiency. We validate our model on multiple dynamic system datasets and compare it with several state-of-the-art methods. Our results demonstrate that MMSGN delivers superior accuracy and mesh quality, excels in managing long-range effects, and maintains high computational efficiency. Furthermore, MMSGN exhibits strong generalization capabilities, scaling effectively to larger physical domains. These advantages make MMSGN well-suited for simulating complex, large-scale dynamic systems across a variety of scenarios. Codes and data will be made publicly accessible upon acceptance.

## 1 Introduction

Numerically solving partial differential equations (PDEs) to model dynamic systems is fundamental in science and engineering but is often computationally intensive, especially in time-sensitive applications requiring rapid inference. This has prompted increased attention across various scientific disciplines, ranging from solid mechanics (Haghighat et al., 2021; Bai et al., 2023) to quantum physics (Sellier et al., 2019; Chen et al., 2022), towards the adoption of learning-based surrogate models. These models aim to expedite numerical simulations, addressing the computational challenges associated with traditional solvers.

In fact, dynamic systems experience continuous and sequential evolution over time, with interconnected elements or components displaying high correlations (corresponds to ***Micro-level Info Exchange*** in Figure 1). Such temporal and spatial coherence is crucial and challenging for accurate simulations (Li et al., 2009; Cubuk et al., 2017; Wiewel et al., 2019). In addition, when interaction happens within dynamic systems, it also exhibit long-range effects, where interactions are required far away from the near-local neighborhood (corresponds to ***Macro-level Info Exchange*** in Figure 1). Examples include ocean waves (Booij & Holthuijsen, 1987), atmospheric convection (Emanuel, 1994), structural vibrations (Fahy, 2007), and seismic waves (Kennett, 2009). Notably, highly correlated regions may not be adjacent but behave similarly due to geometric symmetries or dynamic propagation (Lamb & Roberts, 1998; Reistad et al., 2016; Joubaneh & Ma, 2022).

Mesh-based Graph Neural Network (GNN) methods (Sanchez-Gonzalez et al., 2018; Belbute-Peres et al., 2020; Pfaff et al., 2020) have succeeded in simulating dynamic systems on unstructured meshes but often struggle to balance accuracy and computational efficiency,

especially with large datasets or complex structures. Their reliance on stacking multiple message-passing layers can cause over-smoothing (Chen et al., 2020; Yang et al., 2020) and increase computational cost (Fortunato et al., 2022; Cao et al., 2023), highlighting the need for improved models that efficiently capture both local and global dynamics. To mitigate over-smoothing, an emerging approach simulates sub-level mesh graphs to reduce message-passing steps. Methods generating coarser-level sub-graphs based on spatial proximity (Liu et al., 2021; Janny et al., 2023), manual coarsening (Fortunato et al., 2022), or random pooling (Li et al., 2020) face drawbacks like manual effort or inaccurate mesh edges. Recent advancements (Cao et al., 2023) automate coarsening using

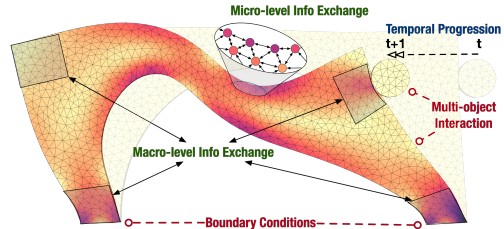

Figure 1: Visualization of hierarchical information exchange, occurring at both micro- and macro-levels, throughout a solid mechanics simulation from time $t$ to $t+1$. In this scenario, a rigid object is interacting with a hyperelastic beam that is anchored to the ground, resulting in deformation.

topological mesh information, improving efficiency. However, they often introduce edges lacking spatial significance and overlook prior information like boundary conditions or material properties (Figure 1), which is crucial for message propagation. In fact, by designing models that align with the natural behavior of real-world dynamic systems, we can optimize the simulation process by effectively capturing the hierarchical interactions inherent in physical phenomena. This alignment not only boosts computational efficiency by minimizing redundant calculations but also improves accuracy by ensuring that information exchange closely reflects actual physical processes.

In this paper, we propose a model that captures both short-range and long-range interactions while balancing accuracy and computational efficiency. By incorporating prior knowledge of real-world systems, we introduce a physically aligned hierarchical information exchange pipeline integrating micro-level and macro-level exchanges. Our Mesh-based Multi-Segment Graph Network (MMSGN) combines local interactions with global exchanges, aligning the mesh hierarchy with physical properties to seamlessly capture dynamics. Evaluations on public datasets (*CylinderFlow* and *Deforming-Plate* (Pfaff et al., 2020)) and a newly developed *DeformingBeam* dataset demonstrate consistently superior performance across various metrics. The main contributions of this paper are summarized as follows:

- We introduce a novel framework that integrates micro-level local interactions with macro-level global exchanges, aligning mesh structures with the system's physical properties to accurately capture both local and long-range dynamics.

- We demonstrate that MMSGN achieves high prediction accuracy and superior mesh quality while maintaining computational efficiency, effectively addressing the common trade-off present in existing methods.

- We present the DeformingBeam dataset and its scaled-up version, providing a comprehensive framework for evaluating mesh-based simulation models; using this dataset, we show that MMSGN demonstrates strong scalability to larger and more complex domains.

## 2 RELATED WORKS

### 2.1 GNNS FOR DYNAMIC SYSTEM SIMULATION

The application of Graph Neural Networks (GNN) for dynamic system prediction is an emerging research area in scientific machine learning due to their versatility and effectiveness (Belbute-Peres et al., 2020; Rubanova et al., 2021; Mrowca et al., 2018). Unlike image-based learning methods such as Convolutional Neural Networks (CNNs) (Um et al., 2018; Ummenhofer et al., 2019), GNNs can directly handle unstructured simulation meshes, making them well-suited for simulating systems with complex domain boundaries while ensuring spatial invariance and locality (Battaglia et al., 2018; Wu et al., 2020). The initial application of GNNs to physics-based simulations focused on deformable solids and fluids, with MeshGraphNets (MGN) being a pioneering work in this area (Pfaff et al., 2020). MGN employs a message passing network to learn the dynamics of physical systems.

Building on this foundation, various MGN variants have been proposed: integrating GNNs with Physics-Informed Neural Networks (PINNs) (Gao et al., 2022), enabling long-term predictions by combining GraphAutoEncoder (GAE) and Transformer models (Han et al., 2022), directly predicting steady states through multi-layer readouts (Harsch & Riedelbauch, 2021), generating initial guesses to speed up the convergence of iterative solvers (Shao et al., 2021), and accelerating fine-level simulations by using up-sampled coarse results inferred by GNNs (Belbute-Peres et al., 2020).

## 2.2 HIERARCHICAL MODELS IN GNNS FOR LONG-RANGE DYNAMIC PROPAGATION

For industrial applications such as furnace and aerodynamic simulation, dynamic systems are usually high-dimensional, and some require fine meshes. This can significantly increase computational costs, and as the mesh becomes finer, GNNs must perform more message passing steps to propagate long-range dynamics, leading to accuracy reduction due to over-smoothing (Li et al., 2018). To address this issue, several hierarchical models have been introduced recently. These hierarchical models can be categorized into two types. The first type includes dual-level structures. For instance, GMR-GMUS (Han et al., 2022) utilizes a pooling method to select pivotal nodes through uniform sampling. Similarly, the EAGLE (Janny et al., 2023) employs a clustering-based pooling method along with transformer mechanism, showing promising performance in fluid dynamics. MS-MGN (Fortunato et al., 2022) proposes a dual-layer framework that passes messages at both fine and coarse resolutions for mesh-based simulation learning. The second type encompasses multi-level structures. One such model, BSMS-GNN (Cao et al., 2023), analyzes limitations of existing pooling strategies and introduces a bi-stride pooling method using breadth-first search (BFS) to select nodes. (Yu et al., 2023) propose a similar hierarchical structure as (Cao et al., 2023) but with two different transformers to enable long-range interactions.

## 3 METHODOLOGY

### 3.1 PROBLEM DEFINITION

Let $G = (\mathcal{V}, \mathcal{E})$ be a mesh graph with $\mathcal{V}$ being the set of nodes and $\mathcal{E}$ being the set of edges. The graph has $N = |\mathcal{V}|$ nodes and $E = |\mathcal{E}|$ edges, with adjacency matrix $A \in \mathbb{R}^{N \times N}$ represents graph connectivity. The dynamic system simulation task is to learn a forward model of the dynamic quantities of the mesh graph at next time step $\hat{G}_{t+1}$ given the current mesh graph $G_t$ and (optionally) a history of previous mesh graphs $\{G_{t-1}, \ldots, G_{t-h}\}$. Finally, the rollout trajectory can be generated through the simulator iteratively based on the previous prediction: $G_t, \hat{G}_{t+1}, \ldots, \hat{G}_{t+T}$, where $T$ is the total simulation steps. In this paper, the proposed model (MMSGN) can simulate both Eulerian and Lagrangian systems (Bontempi & Faravelli, 1998). In Eulerian systems, which model the evolution of continuous fields like velocity over a fixed mesh, the graph $\mathcal{E}$ includes only mesh-related edges $\mathcal{E}^M$. Conversely, in Lagrangian systems, where the mesh represents a moving and deforming surface or volume, additional world edges $\mathcal{E}^W$ are incorporated into the graph. These edges enable the model to learn external dynamics such as collision and contact. The node features of node $i$ are denoted by $\mathbf{x}_i$, while the features for an edge between node $i$ and $j$ are indicated by $\mathbf{e}_{ij}$.

### 3.2 HIERARCHICAL INFORMATION EXCHANGE ARCHITECTURE

**Micro-level Information Exchange** In the micro-level or node-level information exchange stage, each node engages in the exchange of information with its neighboring nodes. This process holds particular significance in dynamic systems, where the behavior of adjacent nodes is closely intertwined (Booij & Holthuijsen, 1987; Emanuel, 1994; Fahy, 2007; Kennett, 2009). Furthermore, the micro-level exchange module serves a crucial role in addressing discontinuities that may arise at the boundaries of adjacent mesh segments (Lai et al., 2009). By prioritizing micro-level information exchange, we effectively mitigate discontinuities introduced by subsequent macro-level operations. For detailed information, refer to Section 3.3.1.

**Macro-level Information Exchange** As discussed in Section 1, to avoid the uninterpretable and potentially erroneous dynamics that coarsened graphs or added edges might introduce, we propose preserving the original mesh structure and facilitating long-range information exchange through communication between segmented mesh graphs. While the concept of segmenting meshes is inspired

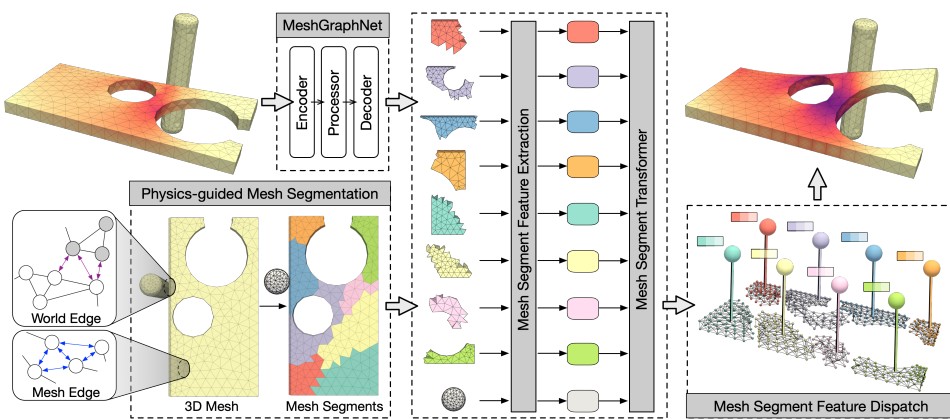

Figure 2: Mesh-based Multi-Segment Graph Network (MMSGN)

by domain decomposition (Toselli & Widlund, 2004), our objectives differ. Traditional decomposition aims to enable parallel computing on each segment to improve computational efficiency, requiring each segment to have a similar number of nodes (Dolean et al., 2015). In contrast, we focus on optimizing mesh segmentation to encourage indirect communication between non-neighboring but related regions for long-range dynamic propagation. Therefore, our approach advocates for segmentation guided by physical properties of dynamic systems. For instance, grouping elements with similar material properties can enhance model convergence by minimizing discontinuities between different materials within each segment (Diao et al., 2023). Additionally, grouping nodes with similar physical behavior can simplify the learning process of the simulation and ensure that similar physical interactions are handled uniformly (Dolean et al., 2015). Details on macro-level information exchange can be found in Section 3.3.2, 3.3.3, and 3.3.4.

## 3.3 MESH-BASED MULTI-SEGMENT GRAPH NETWORK (MMSGN)

### 3.3.1 MESH-BASED GRAPH NEURAL NETWORKS

We adopt the Encoder-Process-Decoder (EPD) (Pfaff et al., 2020) network structure for our micro-level information exchange as it has shown superior performance in dealing with mesh-based graphs. For a given graph $G_t$ at time $t$, the model begins with extracting node and edge features through two separate Multi-Layer Perceptrons (MLPs):

$$\mathbf{h}_{i,t}^0 = f_n(\mathbf{x}_{i,t}), \quad \mathbf{h}_{ij,t}^{M,0} = f_e^M(\mathbf{e}_{ij,t}^M), \quad \mathbf{h}_{ij,t}^{W,0} = f_e^W(\mathbf{e}_{ij,t}^W), \tag{1}$$

where $\mathbf{x}_{i,t}, \mathbf{e}_{ij,t}^M \in \mathcal{E}^M$, and $\mathbf{e}_{ij,t}^W \in \mathcal{E}^W$ denote node feature, mesh edge feature, and world edge feature vector at time $t$, respectively. For Lagrangian systems, world edges are created by spatial proximity, where for a fixed radius $r_W$, a world edge is added between nodes $i$ and $j$ when $|\mathbf{x}_i - \mathbf{x}_j| < r_W$, excluding node pairs already connected in the mesh. The outputs of two MLPs (i.e. $f_n$ and $f_e$) for node and edge are denoted as $\mathbf{h}_{i,t}^0$ and $\mathbf{h}_{ij,t}^0$, respectively. Then, a $L$-step message passing (MP) is performed such that each node can receive and aggregate information from neighboring nodes within $L$ steps of edge traversing. For each MP from 1 to $L$, the node and edge representations are updated as:

$$\mathbf{h}_{i,t}^l = f_n^l(\mathbf{h}_{i,t}^{l-1}, \sum_{j \in Adj(i)} \mathbf{h}_{ij,t}^{M,l-1}, \sum_{j \in Adj(i)} \mathbf{h}_{ij,t}^{W,l-1}),$$

$$\mathbf{h}_{ij,t}^{M,l} = f_e^l(\mathbf{h}_{ij,t}^{M,l-1}, \mathbf{h}_{i,t}^{l-1}, \mathbf{h}_{j,t}^{l-1}), \quad \mathbf{h}_{ij,t}^{W,l} = f_e^l(\mathbf{h}_{ij,t}^{W,l-1}, \mathbf{h}_{i,t}^{l-1}, \mathbf{h}_{j,t}^{l-1}), \tag{2}$$

where $Adj(i)$ denotes all adjacent nodes of node $i$. Up until this point, the node and edge information of the graph $G_t$ are updated. Additionally, we implement a technique from (Godwin et al., 2021), which involves corrupting the input graph with noise and adding a noise-correcting node-level loss. We evaluate the impact of varying the number of message passing steps during micro-level information exchange step, where details can be found in Appendix E.1.

### 3.3.2 Physics-guided Mesh Segmentation

Our physics-guided mesh segmentation approach is designed to create high-quality mesh segments that capture both geometric and physical properties of dynamic systems. By leveraging prior physical information, the method ensures that mesh segments reflect the system's physical behavior, enabling efficient and accurate simulations.

**Mathematical Notation** – We define the segmentation policy $\pi(G) = f_s(G, I)$, where the segmentation function $f_s$ takes the input graph $G$ and prior physical information $I$ (e.g., boundary conditions, material properties), and outputs a set of graph segments $\{S_1^0, S_2^0 \ldots, S_K^0\}$. The superscript 0 denotes non-overlapping segmentation. For each segment $S_k = (\mathcal{V}_{S_k}, \mathcal{E}_{S_k})$, the set of nodes $\mathcal{V}_{S_k} \subseteq \mathcal{V}$ and $\mathcal{E}_{S_k} \subseteq \mathcal{E}$ are subsets of the original graph $G$. The union of all segments reconstructs the original graph, such that $\mathcal{V} = \cup \mathcal{V}_{S_k}^0$ and $\mathcal{E} = \cup \mathcal{E}_{S_k}^0$.

In some cases, it may be beneficial to allow for overlapping segments, where nodes in $\mathcal{V}$ can belong to more than one segment. This overlap helps create smoother transitions between segments and reduces discontinuities at segment boundaries. We define the overlap amount by $\delta \in \mathbb{N}$, with $\delta = 0$ representing no overlap. For $\delta > 0$, the node set $\mathcal{V}_{S_k}^\delta$ is defined recursively as as $\mathcal{V}_{S_k}^\delta = \mathcal{V}_{S_k}^{\delta-1} \cup \{Adj(i) \mid i \in \mathcal{V}_{S_k}^{\delta-1}\}$. To simplify the presentation, we disregard the superscript $\delta$ in the remainder of this paper. The effect of adding overlapping segments is discussed in our ablation study, as shown in Table 5.

**Hybrid Segmentation Overview** – Traditional graph segmentation methods (Alpert & Yao, 1995; Karypis & Kumar, 1998; Delingette, 1999) primarily focus on partitioning the graph based on geometric properties and computational efficiency, often without considering the underlying physical properties of the data. On the other hand, superpixel methods (Vedaldi & Soatto, 2008; Veksler et al., 2010; Achanta et al., 2012) in computer vision aim to segment images into non-overlapping regions to simplify image processing tasks by grouping pixels with user-defined similarity measures. However, this process requires initializing reasonable cluster centers in the image before clustering, which plays an important role in segmentation quality and spatial coherence. To harness the strengths of both domains, we integrate a *graph-based segmentation method* ($f_{gb}$) for generating initial mesh segments and a *superpixel-based segmentation method* ($f_{sb}$) to refine these partitions using self-defined features based on prior information $\mathcal{I}$. This hybrid approach combines the geometric efficiency of graph-based methods with the adaptive feature-based refinement of superpixel methods, resulting in mesh graph segments that are both high-quality and adaptable to diverse dynamic systems.

To be more specific, we first use METIS (Karypis & Kumar, 1998) for initial mesh segmentation due to its great balance of partition quality and speed. Formally, given a graph $G$, the partition function $f_{gb}$ will split it into $K$ non-overlapped mesh-segment graphs: $\{S_1, \ldots, S_K \mid S_i \cap S_j = \varnothing, \forall i \neq j\} = f_{gb}(G)$. Then, we apply SLIC (Achanta et al., 2012), the state-of-the-art superpixel-based clustering methods, to these mesh segments to iteratively update the segmentation centroids $\{C_1, \ldots, C_K\}$ and corresponding node assignments using prior information $\mathcal{I}$.

**Physics-guided Segmentation** – For node $i$ in graph $G$, we represent it by its spatial coordinates $\mathbf{x}_i$, its shortest distance to obstacle nodes $d_i^{obs}$, and its shortest distance to boundary nodes $d_i^{bd}$. We use a self-defined function to convert distances $d_i^{obs}, d_i^{bd}$ to features $f_i^{obs}, f_i^{bd}$. For a given mesh segment $S_k$ containing $|\mathcal{V}_{S_k}|$ nodes, we define its centroid $C_k$ as its mean value along the features:

$$C_k = [\mathbf{x}_{C_k}, f_{C_k}^{obs}, f_{C_k}^{bd}]^T = \frac{1}{|\mathcal{V}_{S_k}|} \sum_{i \in \mathcal{V}_{S_k}} [\mathbf{x}_i, f_i^{obs}, f_i^{bd}]^T. \tag{3}$$

Within each iteration, we improve the mesh segmentation by minimizing a distance measure that considers both physical similarity and spatial proximity. The distance measure $d(i, C_k)$ between a node $i \in \mathcal{V}$ and a segment's centroid $C_k$ is defined as:

$$d(i, C_k) = \|f_i^{obs} - f_{C_k}^{obs}\| + \|f_i^{bd} - f_{C_k}^{bd}\| + m\|\mathbf{x}_i - \mathbf{x}_{C_k}\|, \tag{4}$$

where $m$ is used to control the compactness of a mesh segment.

The pseudo code of our proposed physics-guided mesh segmentation method can be found in Algorithm 1. In Appendix C, we present a comprehensive comparison of various segmentation methods and their variants based on different distance measures. Additionally, we evaluate the impact of varying the number of mesh segments on model performance in Appendix E.2 and Appendix F.2.

We also introduce several metrics to measure quality of different mesh segmentation, specifically to understand the intra-segment and inter-segment characteristics, which can be found in Appendix D.2

### 3.3.3 MESH SEGMENT FEATURE EXTRACTION

**Segment Encoding (SE)** – In order to extract a global feature for each mesh segment, we perform average pooling on all node vectors in $S_k$ and apply a MLP ($f_s$) to get the fixed-sized segment embedding:

$$\mathbf{h}_{S_k,t} = f_s\left(\frac{1}{|\mathcal{V}_{S_k}|} \sum_{i \in \mathcal{V}_{S_k}} \mathbf{h}_{i,t}^L\right). \tag{5}$$

**Position Encoding (PE)** – As dynamic effect propagates continuously over mesh domains, knowing relative location among segments could provide extra information for next-step macro-level information exchange and increase expressivity of the network. Mathematically, for each pair of mesh segment graph, $\{S_i, S_j\}$, their relative positional information can be obtained through segment-level adjacency matrix $A^K \in \mathbb{R}^{K \times K}$:

$$A_{S_i S_j}^K = |\mathcal{V}_{S_i} \cap \mathcal{V}_{S_j}| = \text{Cut}(\mathcal{V}_{S_i}, \mathcal{V}_{S_j}) = \sum_{m \in \mathcal{V}_{S_i}} \sum_{n \in \mathcal{V}_{S_j}} A_{mn}, \tag{6}$$

where $\text{Cut}(\cdot)$ is a graph operator that counts the number of connection edges between node clusters in mesh segment graph $S_i$ and $S_j$. The PE for the $k$-th segment, denoted as $\mathbf{p}_{S_k,t}$, is processed through an MLP layer ($f_{sp}$) and then added to update the SE from Eq (5) as follows: $\mathbf{h}_{S_k,t} \leftarrow \mathbf{h}_{S_k,t} + f_{sp}(\mathbf{p}_{S_k,t})$.

We can further enhance the network's expressivity by adding absolute PE to the graph nodes. We use an MLP ($f_{np}$) to process each node's PE ($\mathbf{p}_{i,t}$), calculated via random-walk structural encoding (RWSE) (Dwivedi et al., 2021), and add this to the input node feature. Thus, Eq (1) becomes $\mathbf{h}_{i,t}^0 = f_n(\mathbf{x}_{i,t} + f_{np}(\mathbf{p}_{i,t}))$. By incorporating node PE directly into the input features, these features participate in the micro-level information exchange described in Section 3.3.1, potentially improving the continuity of the extracted mesh segment features. Table 6 presents ablation studies show how adding or omitting PE affects prediction results.

### 3.3.4 MESH SEGMENT TRANSFORMER

We construct a fully connected mesh segment graph, where the $i$-th mesh segment feature is represented by $\mathbf{h}_{S_i}$. Note that since the transformer operates on mesh segments rather than individual mesh nodes, and the total number of mesh segments ($K$) is significantly smaller than the total number of mesh nodes ($N$), the computational cost of our transformer is substantially reduced compared to a traditional graph transformer that operates on graph nodes (i.e. $O(K^2) \ll O(N^2)$). The $l$-th block of the mesh segment transformer layer is defined as follows:

$$\mathbf{a}_{S_i S_j}^{k,l} = \text{softmax}_{S_j}\left(\frac{\mathbf{Q}_h^{k,l} \text{LN}(\mathbf{h}_{S_i}^l) \cdot \mathbf{K}_h^{k,l} \text{LN}(\mathbf{h}_{S_j}^l)}{\sqrt{d_h}}\right), \tag{7}$$

$$\bar{\mathbf{h}}_{S_i}^l = \|_{k=1}^H \sum_{j=1}^K \mathbf{a}_{S_i S_j}^{k,l} (\mathbf{V}_h^{k,l} \text{LN}(\mathbf{h}_{S_j}^l)), \tag{8}$$

$$\mathbf{h}_{S_i}^{l+1} = \mathbf{h}_{S_i}^l + \mathbf{O}_h^l \bar{\mathbf{h}}_{S_i}^l + \text{FFN}_h^l(\text{LN}(\mathbf{h}_{S_i}^l + \mathbf{O}_h^l \bar{\mathbf{h}}_{S_i}^l)), \tag{9}$$

where $\mathbf{a}_{S_i S_j}^{k,l}$ is self-attention weight between $S_i$ and $S_j$. $\mathbf{Q}_h^{k,l}, \mathbf{K}_h^{k,l}, \mathbf{V}_h^{k,l} \in \mathbb{R}^{d_h \times d}$ are trainable parameters, and $\mathbf{O}_h^l \in \mathbb{R}^{d \times d}$ is the learned output project matrix. $k = 1$ to $H$ denotes the number of attention heads, and $\|$ denotes concatenation. $d_h$ is the dimension of mesh segment feature for each head, and $d$ is the input and output dimension. We adopt a Pre-Layer Norm architecture (Xiong et al., 2020), which is denoted as $\text{LN}(\cdot)$, and the point-wise Feed Forward Network is represented as $\text{FFN}(\cdot)$. The mesh segment transformer module facilitates information exchange among all mesh segments, updating the feature of each segment $\mathbf{h}_{S_i}$ after passing through $L_S$ mesh segment transformer blocks.

### 3.3.5 MESH SEGMENT FEATURE DISPATCH AND TRAINING

The mesh segment feature dispatch module (as shown in Figure 2) integrates information obtained from both macro-level and micro-level exchanges. Specifically, the final feature for node $i$ at time

step $t$ is updated as $\mathbf{h}_{i,t} \leftarrow [\mathbf{h}_{i,t}, \mathbf{h}_{S_i,t}]$ where $i \in \mathcal{V}_{S_i}$. This ensures that each node incorporates information from both neighboring mesh nodes and spatially distant, yet correlated regions. Finally, we train our dynamics model by supervising on the per-node output features $\hat{\mathbf{x}}_{i,t+1}$, produced by feeding $\mathbf{h}_{i,t}$ into a MLP-based decoder, using a $L_2$ loss between $\hat{\mathbf{x}}_{i,t+1}$ and the corresponding ground truth values $\mathbf{x}_{i,t+1}$.

# 4 EXPERIMENT

## 4.1 EXPERIMENT SETUP

**Datasets** – We use two public datasets from (Pfaff et al., 2020): *CylinderFlow* (fluid flows around a cylinder) and *DeformingPlate* (elastic plate deformed by an actuator). We also create a new dataset, *DeformingBeam*, featuring a hyperelastic beam deformed by an actuator in a 3D mesh. Details of the datasets can be found in the appendix A. We create *DeformingBeam* dataset for three major reasons: (1) This dataset exhibits long-range interactions, with the largest graph diameter compared to the other two datasets (Table 2). (2) The inclusion of diverse mesh structures significantly increases the complexity of the underlying physics, making the task more challenging. (3) The dataset allows for the generation of directly scaled-up versions, enabling comprehensive generalization tests.

**MMSGN and Baselines** – As a default configuration for our MMSGN model, we use 7 message passing steps in the mesh graph network. The mesh segment transformer adopts 4 self-attention layers with 8 heads. We compare our method to five baseline models: 1) *GCN* (Kipf & Welling, 2016; Belbute-Peres et al., 2020), a basic GNN structure widely used for simulating fluid dynamics; 2) *g-U-Nets* (Gao & Ji, 2019; Alsentzer et al., 2020), a representative method that incorporates graph pooling modules to enhance long-range interactions; 3) *MeshGraphNets* (MGNs) (Pfaff et al., 2020), a single-level GNN architecture that achieves exceptional performance and generalizability across various dynamic systems; 4) *BSMS-GNN* (Cao et al., 2023), a recent work featuring a multi-level hierarchical GNN architecture that aims to enhance computational efficiency in simulating physical systems; and 5) *EAGLE*(Janny et al., 2023), a recent work presenting a clustering-based pooling method along with transformer to enhance performance on large-scale turbulent fluid dynamics. Detailed descriptions of the these models and training procedures can be found in Appendix B.

**Metrics** – In addition to traditional accuracy metrics, we introduce mesh quality metrics to assess the integrity of the predicted mesh in Lagrangian systems, where the mesh moves with the material. Maintaining mesh quality is crucial in these systems because changes in mesh elements over time can lead to numerical errors and misrepresentation of dynamic behaviors. Conversely, in Eulerian systems with a fixed mesh, mesh quality is less critical since the mesh remains static. We define two mesh quality metrics: Geometric Fidelity (GF), which measures how well the with the predicted node positions conforms to the system's true geometry, and Mesh Continuity (MC), which evaluates the uniformity of predicted mesh cell sizes to ensure stability. These metrics are mathematically defined as follows:

$$\text{GF} = \max\Big\{h(\mathcal{V}, \hat{\mathcal{V}}), h(\hat{\mathcal{V}}, \mathcal{V})\Big\} \quad \text{and} \quad \text{MC} = \frac{1}{C}\sum_{i=1}^{C}\frac{\max_{c_j \in \text{Adj}(c_i)} V(c_j)}{\min_{c_j \in \text{Adj}(c_i)} V(c_j)}, \tag{10}$$

where $h(\mathcal{V}, \hat{\mathcal{V}}) = \sup_{\mathbf{x} \in \mathcal{V}} \inf_{\hat{\mathbf{x}} \in \hat{\mathcal{V}}} \|\mathbf{x} - \hat{\mathbf{x}}\|$ is the directed Hausdorff distance (Huttenlocher et al., 1993) from the ground-truth node set $\mathcal{V}$ to the predicted node set $\hat{\mathcal{V}}$, $\text{Adj}(c_i)$ is the neighboring cells of cell $c_i$, and $V(c_i)$ calculates the volumetric area for $c_i$. To achieve a more holistic assessment of the predicted meshes, two additional metrics are used (Wu et al., 2021; Zienkiewicz & Taylor, 2005) for mesh quality evaluation (available in Appendix D.1).

## 4.2 RESULTS AND DISCUSSION

### 4.2.1 OUTSTANDING PERFORMANCE OF MMSGN ACROSS MULTIPLE DATASETS

The results in Table 1 demonstrate the superior performance of our MMSGN model compared to other baselines across various evaluation metrics. Specifically, for the CylinderFlow dataset, MMSGN achieves a remarkable 36% reduction in test RMSE-all compared to the second-best performing model, EAGLE. This improvement is even more pronounced for the DeformingPlate dataset, where

Table 1: Comparison of results with state-of-the-art methods across three datasets, where each model is trained independently for each dataset. Prediction accuracy is evaluated using Root Mean Square Error (RMSE), with the output being the 2D velocity field for CylinderFlow and the 3D position for DeformingBeam and DeformingPlate. Errors are reported for 1-step rollout, 50-step rollouts, and the entire trajectory. Mesh quality is assessed using Geometric Fidelity (GF) and Mesh Continuity (MC). Results are averaged over three experiments with different random seeds and presented as mean and standard deviation. Additional evaluations of these methods using extended metrics are presented in Table 3.

| DATASET | MODEL | Mesh Quality Metrics | | Prediction Error Metrics | | |
|---|---|---|---|---|---|---|
| | | GF ($\times 10^{-3}$)↓ | MC ($\times 10^{-3}$)↓ | RMSE-1 ($\times 10^{-5}$) | RMSE-50 ($\times 10^{-4}$) | RMSE-all ($\times 10^{-4}$) |
| CYLINDER FLOW | GCN | - | - | $764 \pm 32$ | $425 \pm 82$ | $1887 \pm 358$ |
| | $g$-U-NET | - | - | $423 \pm 4$ | $199 \pm 37$ | $843 \pm 141$ |
| | MGN | - | - | $274 \pm 15$ | $64.4 \pm 3.4$ | $481 \pm 53$ |
| | BSMS-GNN | - | - | $\mathbf{202 \pm 24}$ | $280 \pm 9$ | $1373 \pm 90$ |
| | EAGLE | - | - | $507 \pm 25$ | $71.5 \pm 3.2$ | $583 \pm 29$ |
| | MMSGN (OURS) | - | - | $320 \pm 29$ | $\mathbf{63.6 \pm 2.6}$ | $\mathbf{372 \pm 27}$ |
| DEFORMING PLATE | GCN | $24.0 \pm 0.6$ | $11.0 \pm 0.3$ | $34.8 \pm 0.6$ | $26.1 \pm 0.1$ | $169 \pm 1$ |
| | $g$-U-NET | $36.1 \pm 8.5$ | $20.1 \pm 0.5$ | $41.2 \pm 0.2$ | $30.4 \pm 0.8$ | $179 \pm 7$ |
| | MGN | $12.7 \pm 0.9$ | $9.25 \pm 0.39$ | $\mathbf{22.8 \pm 0.2}$ | $20.0 \pm 0.4$ | $147 \pm 3$ |
| | BSMS-GNN | $23.8 \pm 2.6$ | $18.3 \pm 4.4$ | $30.3 \pm 5.6$ | $23.7 \pm 3.5$ | $118 \pm 4$ |
| | EAGLE | $6.75 \pm 0.8$ | $5.56 \pm 0.12$ | $36.4 \pm 5.2$ | $5.63 \pm 1.7$ | $38.7 \pm 1.8$ |
| | MMSGN (OURS) | $\mathbf{4.43 \pm 0.08}$ | $\mathbf{4.78 \pm 0.09}$ | $26.9 \pm 0.3$ | $\mathbf{3.10 \pm 0.08}$ | $\mathbf{26.0 \pm 1.3}$ |
| DEFORMING BEAM | GCN | $4.91 \pm 0.36$ | $54.8 \pm 8.2$ | $7.25 \pm 0.12$ | $5.08 \pm 0.11$ | $30.7 \pm 4.1$ |
| | $g$-U-NET | $4.91 \pm 0.50$ | $34.7 \pm 1.8$ | $7.28 \pm 0.39$ | $5.09 \pm 0.23$ | $31.7 \pm 4.0$ |
| | MGN | $0.82 \pm 0.04$ | $16.9 \pm 0.1$ | $4.43 \pm 0.08$ | $2.41 \pm 0.16$ | $4.72 \pm 0.27$ |
| | BSMS-GNN | $0.99 \pm 0.03$ | $32.5 \pm 0.5$ | $6.86 \pm 0.09$ | $1.95 \pm 0.22$ | $4.98 \pm 0.71$ |
| | EAGLE | $0.64 \pm 0.04$ | $5.98 \pm 0.43$ | $1.51 \pm 0.04$ | $0.67 \pm 0.12$ | $4.22 \pm 0.30$ |
| | MMSGN (OURS) | $\mathbf{0.33 \pm 0.01}$ | $\mathbf{5.24 \pm 0.04}$ | $\mathbf{1.18 \pm 0.01}$ | $\mathbf{0.33 \pm 0.02}$ | $\mathbf{2.07 \pm 0.15}$ |

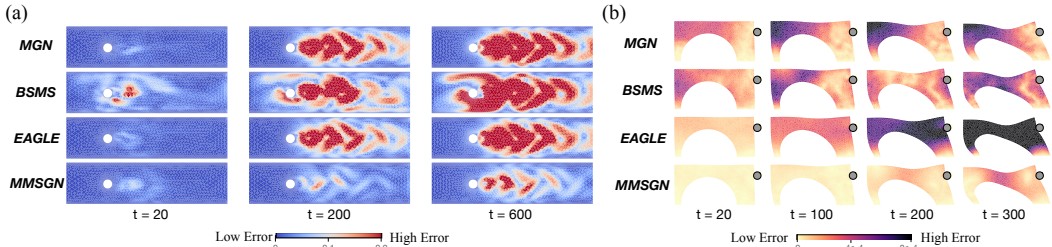

Figure 3: Comparison of simulation results for different models across two datasets: CylinderFlow (left) and DeformingBeam (right). Plots are color-coded by RMSE error over $t$-step rollouts.

MMSGN reduces the test RMSE-all by 42%. Similarly, for DeformingBeam dataset, MMSGN demonstrates a 51% reduction in test RMSE-all. Such exceptional performance in 50-step and longer-step predictions underscores its enhanced capability for long-term predictions. In addition to achieving superior prediction accuracy, MMSGN demonstrates excellent mesh quality, with up to a 48% reduction in GF and a 14% reduction in MC compared to the second-best model across both Lagrangian system datasets.

Additionally, Figure 3 presents two selected test cases for rollout visualization, highlighting that MMSGN achieves notably lower RMSE errors in areas where other methods struggle, particularly at later time steps and in regions further from the inlet or contact point. Overall, our MMSGN model consistently outperforms baseline models across all datasets and evaluation metrics, solidifying its effectiveness in modeling challenging dynamic systems. Additional simulation results can be found in Appendix H.

### 4.2.2 ACHIEVING A BALANCE BETWEEN ACCURACY AND EFFICIENCY

According to Figure 4(a), the MGN model performs well on small-diameter datasets like Cylinder-Flow, effectively capturing short-range effects. However, its performance drops on larger datasets like DeformingPlate and DeformingBeam due to oversmoothing and slow inference caused by excessive message passing and world-edges, reducing overall efficiency. The BSMS model excels in memory efficiency due to its bi-stride pooling, but this comes at the expense of mesh quality and accuracy, as the pooling introduces spatially insignificant edges. It also has slower inference times due to the

complexity of reconstructing fine-grained details and managing long-range dynamics. The EAGLE model performs adequately but struggles with long-range effects due to its graph clustering and pooling methods, which lack physics-informed guidance. While it shows reasonable efficiency in DeformingPlate and DeformingBeam, its computational performance declines dramatically on CylinderFlow due to the increased amount of clusters needed under dense meshes.

MMSGN achieves the largest filled areas across all three datasets, demonstrating high prediction accuracy, superior mesh quality, and strong computational efficiency. This is due to its physically aligned hierarchical structure, which integrates micro-level local interactions with macro-level global exchanges to capture both short- and long-range dynamics. The physics-guided segmentation ensures nodes within segments exhibit similar behaviors, while macro-level exchanges efficiently handle long-range dependencies. This alignment with physical dynamics enables MMSGN to maintain high accuracy without compromising efficiency, making it highly effective for dynamic system simulation. More comprehensive evaluation results can be found in Table 8.

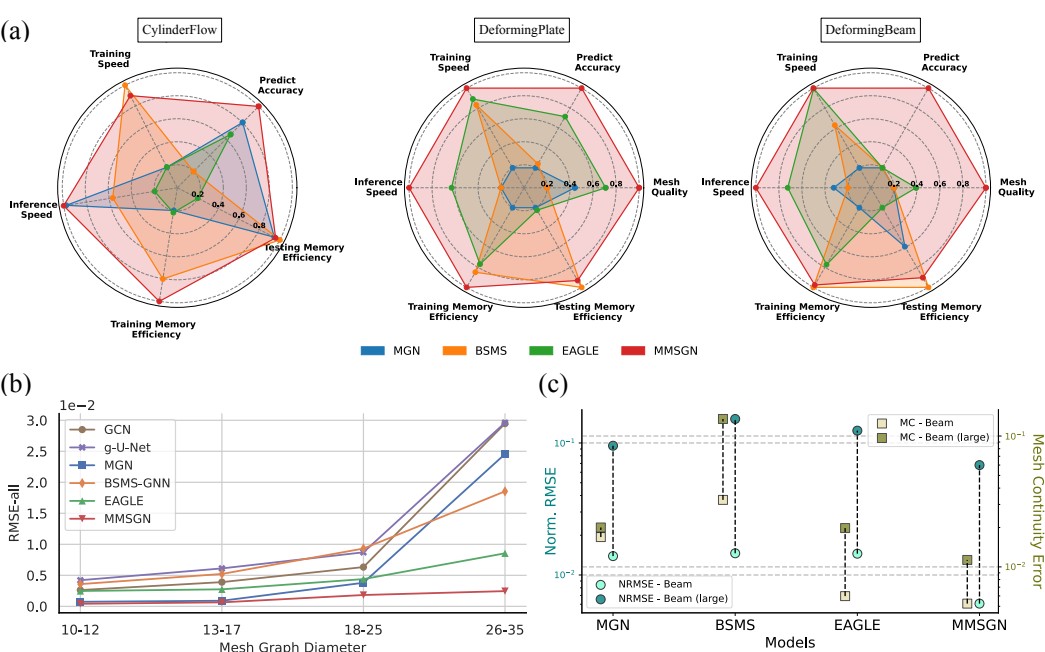

Figure 4: (a) Radar charts comparing the performance of difference models across different metrics under three datasets. For each metric, values are normalized to a 0.2–1.0 scale, where 1.0 represents the best performance. The concentric circles show normalized values from 0.2 (innermost) to 1.0 (outermost). Larger filled areas indicate better overall performance; (b) Illustration of how varying graph diameter impacts prediction accuracy across different models on the DeformingPlate dataset. RMSE-all is averaged over selected cases within a given graph diameter range and across all time steps; (c) Illustration of how each model's prediction accuracy (Normalized RMSE) and mesh quality (MC) change when generalizing from DeformingBeam to its scaled-up version, DeformingBeam (large).

### 4.2.3 Effective Long-Range Dynamics and Scalability to Large Datasets

MMSGN's remarkable performance stems from its ability to effectively handle long-range dynamic effects. In Figure 4(b), the relationship between graph diameter and RMSE-all for the DeformingPlate dataset is shown. While other models experience a significant rise in prediction error as the problem diameter increases, MMSGN only shows a slight increase, demonstrating its superior performance on larger graphs and long-range interactions. Figure 4(c) further illustrates how each model's prediction accuracy (Normalized RMSE) and mesh quality change when generalizing from DeformingBeam to its scaled-up version, DeformingBeam (large). MMSGN consistently achieves the lowest prediction error and mesh continuity error when tested on the larger-scale dataset with a model trained on the

smaller scale. This highlights MMSGN's robust generalization capabilities, making it well-suited for complex, large-scale dynamic systems. Detailed generalization results are presented in Table 7.

### 4.3 ADDITIONAL STUDIES

We conducted additional studies to comprehensively evaluate model performance, hyperparameter selection, and the impact of key architectural designs, with detailed results and discussions provided in the appendices. Metrics for mesh quality and evaluations are presented in Appendix D.1, while segmentation quality metrics and related evaluations are detailed in Appendix D.2. Visualization and discussion of segmentation alignment with system dynamics are included in Appendix D.3. Ablation studies of our approach are discussed in Appendix E, covering the effect of different message-passing steps for micro-level information exchange (Appendix E.1) and the influence of segment extraction methods, segment count, positional encoding, and segment overlap for macro-level information exchange (Appendix E.2). Additionally, a comprehensive analysis of generalization performance is provided in Appendix F, and further insights into computational efficiency are included in Appendix G.

## 5 CONCLUSION

In this paper, we introduced the Mesh-based Multi-Segment Graph Network (MMSGN), a novel approach that enhances dynamic system simulations through a hierarchical information exchange pipeline. Our extensive evaluations demonstrate that MMSGN outperforms traditional models, offering significant improvements in accuracy and computational efficiency, particularly in scenarios involving long-range dynamics and larger physical domains. The adaptability of MMSGN to large-scale graphs underscores its potential for real-world applications in complex physical systems. However, the method has limitations, including the absence of hard constraints on contact meshes, which can result in overlapping meshes, and it has no guarantees on physical consistency at segmentation interfaces. These are important areas for future work to improve the robustness and applicability of the model. We have not observed any negative impacts arising from this work.

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

# A DATASETS

## A.1 DATASETS FOR UNSTRUCTURED MESH-BASED SIMULATIONS

Mesh-based dynamics simulation datasets have been developed as benchmarks to evaluate the performance of proposed models. As a cornerstone in the field, MeshGraphNets Pfaff et al. (2020) introduced a collection of datasets, encompassing cloth simulation, materials deformation and fluid flow, showcasing the versatility of GNNs in various problems involving unstructured mesh simulations. These datasets have been extensively adopted as benchmarks for developing new models. As the field shifts towards tackling more complex and large-scale systems, EAGLE Janny et al. (2023) presented a large-scale fluid dynamics dataset capturing unsteady and turbulent airflows. Similarly, BSMS-GNN Cao et al. (2023) provides the InflatingFont dataset, which focuses on the quasi-static inflation of enclosed elastic surfaces.

To demonstrate our model's generality across diverse dynamics and mesh configurations, we employed the *CylinderFlow* and *DeformingPlate* datasets in this study. These widely-used datasets from MeshGraphNets encompass both Eulerian and Lagrangian systems, providing a comprehensive evaluation of our model's performance across different simulation paradigms. Additionally, we developed the *DeformingBeam* dataset, which features meshes with a large graph diameter and complex long-range interactions spanning distant regions of the mesh. Existing datasets often lack this level of complexity, limiting their effectiveness in testing advanced models. We also generated a scaled-up version of the DeformingBeam dataset, enabling the evaluation of generalization performance from small-scale to large-scale scenarios, an important consideration for industrial-level simulations. The details of the investigated datasets are desbribed below.

## A.2 DATASETS DETAILS

**CylinderFlow** – This public dataset includes simulations of transient incompressible flow around a cylinder, with varying diameters and locations, on a fixed 2D Eulerian mesh. In all fluid domains, the node type distinguishes fluid nodes, wall nodes and inflow/outflow boundary nodes. The inlet boundary conditions are given by a prescribed parabolic profile, $u_{in} = u_0[1 - 4(y/H)]$ where $u_0$ and H are the centerline velocity and the distance between the sidewalls, respectively. The dataset contains 1000 training simulations, 100 validation simulations and 100 test simulations.

**DeformingPlate** – This public dataset includes simulations of hyperelastic plates deformed by a moving obstacle, with variations in plate design and obstacle design. The node types are plate nodes, handle nodes that are fixed and obstacle nodes. This dataset contains 1200 training simulations, 100 validation simulations and 100 test simulations.

**DeformingBeam** – This dataset is generated using *solids4foam* which a toolbox for performing solid mechanics and fluid-solid interaction simulations in OpenFOAM and foam-extend. A nearly incompressible neo-Hookean model is used where the material properties are density $\rho_0 = 1000$ kg/m$^3$, Youngs's modulus $E = 1$ MPa and Poisson's ratio $\nu = 0.4$. The beam comes in different geometries with various initial conditional and boundary conditions. The node types are plate nodes, handle nodes that are fixed and obstacle nodes. This dataset contains 355 training simulations, 40 validation simulations and 60 test simulations.

**DeformingBeam (large)** – A large domain DeformingBeam dataset is created for generalization studies. The physical domain size is doubled. The size of the mesh cell is kept consistent with the regular DeformingBeam dataset. This generalization dataset has 112 simulations.

Table 2: Detailed information for each dataset.

| DATASET | AVG. # NODES | # STEPS | MESH TYPE | GRAPH DIAMETER | NODE FEATURE | EDGE FEATURE | OUTPUT |
|---|---|---|---|---|---|---|---|
| CYLINDERFLOW | 1885 | 600 | TRIANGLE, EULERIAN, 2D | 11 | $\mathbf{v}_i, \mathbf{n}_i$ | $\mathbf{m}_{ij}, |\mathbf{m}_{ij}|$ | $\dot{\mathbf{v}}_i$ |
| DEFORMINGPLATE | 1271 | 400 | TETRAHEDRON, LAGRANGIAN, 3D | $16.9 \pm 5.8$ | $\mathbf{x}_i, \dot{\mathbf{x}}_{\text{OBS}}, \mathbf{n}_i$ | $\mathbf{x}_{ij}, |\mathbf{x}_{ij}|, \mathbf{m}_{ij}, |\mathbf{m}_{ij}|$ | $\dot{\mathbf{x}}_i$ |
| DEFORMINGBEAM | 1542 | 400 | PRISM, LAGRANGIAN, 3D | $41.3 \pm 11.8$ | $\mathbf{x}_i, \dot{\mathbf{x}}_{\text{OBS}}, \mathbf{n}_i$ | $\mathbf{x}_{ij}, |\mathbf{x}_{ij}|, \mathbf{m}_{ij}, |\mathbf{m}_{ij}|$ | $\dot{\mathbf{x}}_i$ |
| DEFORMINGBEAM (LARGE) | 4540 | 400 | PRISM, LAGRANGIAN, 3D | $82.1 \pm 23.0$ | $\mathbf{x}_i, \dot{\mathbf{x}}_{\text{OBS}}, \mathbf{n}_i$ | $\mathbf{x}_{ij}, |\mathbf{x}_{ij}|, \mathbf{m}_{ij}, |\mathbf{m}_{ij}|$ | $\dot{\mathbf{x}}_i$ |

## B  MODEL DETAILS

### B.1  MMSGN

The GNN part of MMSGN adopts the encoder and graph processor in the MGN model Pfaff et al. (2020). The basic building block is Multi-Layer Perceptron (MLP). The MLP has 3 layers, a hidden dimension of 128, ReLU activation and single layer of Layer Normalization at the end. The node encoder and edge encoder(s) are 3-layer MLPs. By default, the MMSGN has 7 message passing steps in the GNN. The mesh segment transformer consists of 4 self-attention layers, each with 8 heads. The output decoder is a 3-layer MLP without Layer Normalization. For DeformingPlate and DeformingBeam, MMSGN only considers world edges between contacting mesh objects. The world edge radius is set to 0.01 for DeformingPlate and 0.002 for DeformingBeam.

### B.2  BASELINES

**GCN** – The GCN model consists of 15 GCN layers with a hidden dimension of 128. The GCN model does not have edge input. Node input includes $\mathbf{x}_i$ for CylinderFlow. The implementation is from PyTorch Geometric.

**g-U-Net** – The g-U-Net model is a modified version from PyTorch Geometric. Instead of GCN layers, it is built using the GNN layers similar to MGN. The level of scale is 7 for CylinderFlow, 6 for DeformingPlate and 4 for DeformingBeam.

**MGN** – Our implementation of MGN follows the one described in Pfaff et al. (2020). The processor of MGN contains 15 MP steps. World edges are constructed as specified in the paper, with a world edge radius of 0.03 for DeformingPlate and 0.003 for DeformingBeam.

**BSMS-GNN** – We followed the BSMS-GNN implementation Cao et al. (2023) from `https://github.com/Eydcao/BSMS-GNN`. We introduced a modification to the original code by incorporating output normalization, which we observed to enhance the model's performance. For CylinderFlow and DeformingPlate, we used the same number of multi-scale levels as specified in the BSMS-GNN paper, at 7 and 6 levels, respectively. The number of multi-scale levels for DeformingBeam is set at 4 as an optimal configuration.

**EAGLE** – The implementation of EAGLE follows the paper Janny et al. (2023) and the code repository `https://github.com/eagle-dataset/EagleMeshTransformer`. We set the number of nodes per cluster at 20, which offers a balanced performance and efficiency according to the paper. This results in 94 clusters for CylinderFlow, 64 for DeformingPlate, and 38 for DeformingBeam. In addition, we add contacting world edges in EAGLE implementation for DeformingPlate and DeformingBeam to improve the performance. The world edges are added the same as in MMSGN.

### B.3  TRAINING DETAILS

During training, random Gaussian noise is added to the spatial node inputs, as described in Pfaff et al. (2020). For CylinderFlow, all models use a noise scale of 0.02. For DeformingPlate, all models use a noise scale of 0.003. For DeformingBeam, EAGLE and MMSGN use a noise scale of 1e-4 and other models use a noise scale of 1e-3.

For GCN, g-U-Net, MGN, EAGLE and MMSGN, we adopt the same training scheme: For Cylinder-Flow and DeformingPlate, we trained the model for 2M steps. The learning rate starts at 1e-4 and exponentially decays to 1e-6 from 1M to 2M steps. For DeformingBeam, we trained the model for 1M steps. The learning rate starts at 1e-4 and exponentially decays to 1e-6 from 500K to 1M steps.

For BSMS-GNN, we adopt the training scheme from the original implementation. Models for CylinderFlow and DeformingPlate were trained for 50 epochs, corresponding to 3.75M and 3M training steps, respectively. DeformingBeam model was trained for 100 epochs, corresponding to 1.775M training steps.

Across all models and datasets, we use a batch size of 8. Experiments were conducted using PyTorch distributed training over two Nvidia Tesla P100 GPUs.

## C  PHYSICS-GUIDED MESH SEGMENTATION DETAILS

In Figure 5, several cases are selected from each dataset to illustrate the difference of each mesh graph segmentation methods. It's worth to note that the graph will be partitioned only once during the training and testing phase for each simulation, and this partitioning will remain consistent across all time steps. This consistency is because the segmentation is based solely on the system's dynamic properties and initial conditions prior to the start of the simulation.

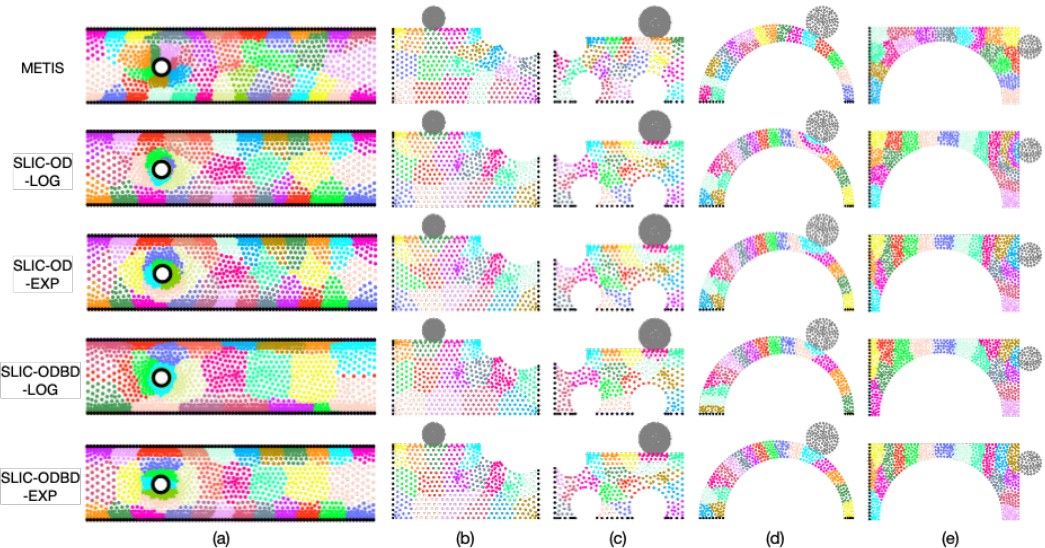

Figure 5: Illustration of different segmentation methods under various cases: (a):CylinderFlow; (b)(c): DeformingPlate; (d)(e): DeformingBeam. Mesh nodes are colored based on segment id and all boundary nodes are colored in black.

METIS (Karypis & Kumar, 1998) is a graph partitioning technique that efficiently divides meshes into approximately equal-sized partitions. It leverages multilevel partitioning algorithms to minimize the edge-cut or communication costs between the resulting partitions. We employ METIS due to its versatility in creating a user-specified number of equal-sized mesh segments. SLIC (Achanta et al., 2012) is a clustering algorithm employed for partitioning data. In our approach, we adapt SLIC to segment the mesh based on physics-informed features. These features could guide SLIC to create a segmentation that captures the underlying physics of the system. The consequent mesh segments can potentially enable efficient macro-level information exchange tailored to the system's dynamics.

Concretely, for each node $i$, we use its shortest distance to obstacle nodes $d_i^{obs}$, and its shortest distance to boundary nodes $d_i^{bd}$ as the physics-informed features. Since we want to let the feature dominates the SLIC distance measure when the $d_i^{obs}$ or $d_i^{bd}$ is small, we encode the distances to features by $f_{\exp}$ or $f_{\log}$, where

$$f_{\exp}(d) = \exp(-d), \quad f_{\log}(d) = \log(d). \tag{11}$$

Depending whether include $d_i^{bd}$ in features or not and the function to convert distance to features, we design 4 variants of SLIC:

- SLIC-OD-LOG: $f_i = f_{\log}(d_i^{obs})$
- SLIC-OD-EXP: $f_i = f_{\exp}(d_i^{obs})$
- SLIC-ODBD-LOG: $f_i = \left[ f_{\log}(d_i^{obs}), f_{\log}(d_i^{bd}) \right]^T$
- SLIC-ODBD-EXP: $f_i = \left[ f_{\exp}(d_i^{obs}), f_{\exp}(d_i^{bd}) \right]^T$

After we have the physics-informed feature, we can apply the SLIC algorithm described in Algorithm 1 to get the mesh node segments. The compact parameter $m$ is set at 1 for CylinderFlow and

---

**Algorithm 1:** Physics-guided Mesh Segmentation

---

**input** Mesh graph $G = (\mathcal{V}, \mathcal{E})$ at time step 0, number of segments $K$, compactness parameter $m$, average cluster size $S$

**output** Mesh node segmentation $\{\mathcal{V}_{S_k}\}_{k=1}^{K}$

1: Compute physics guided feature $f_i$ for each mesh node $i$
2: **Mesh Segment Initialization via Multilevel Graph Partitioning (METIS):**
3: **Coarsening Phase:**
4: $G_{\text{coarse}} \leftarrow G$
5: **while** size of $G_{\text{coarse}}$ is larger than threshold **do**
6:     Combine pairs of connected nodes in $G_{\text{coarse}}$ to form a coarser graph
7:     $G_{\text{coarse}} \leftarrow$ coarsened graph
8: **end while**
9: **Initial Partitioning:**
10: Partition $G_{\text{coarse}}$ into $K$ segments using a standard partitioning method (e.g., spectral partitioning)
11: **Uncoarsening and Refinement Phase:**
12: **while** $G_{\text{coarse}} \neq G$ **do**
13:     Expand $G_{\text{coarse}}$ to the next finer graph $G_{\text{fine}}$
14:     Project partitions onto $G_{\text{fine}}$
15:     Refine the partitioning on $G_{\text{fine}}$ to improve quality
16:     $G_{\text{coarse}} \leftarrow G_{\text{fine}}$
17: **end while**
18: Obtain initial clusters $\{\mathcal{V}_{S_k}\}_{k=1}^{K}$ from the final partitioning, which will be updated next
19: **Physics-informed Mesh Segment Refinement via Superpixel-based Clustering (SLIC):**
20: **repeat**
21:     **for** each mesh segment centroid $C_k$ **do**
22:         Update $C_k$ by averaging over all mesh nodes assigned to it:

$$C_k = [x_{C_k}, f_{C_k}]^T = \frac{1}{|\mathcal{V}_{S_k}|} \sum_{i \in \mathcal{V}_{S_k}} [x_i, f_i]^T$$

         where $\mathcal{V}_{S_k}$ is the set of mesh nodes assigned to segment $S_k$
23:     **end for**
24:     **for** each mesh node $i \in V$ **do**
25:         Compute the distance measure $d(i, C_k)$ to each cluster center $C_k$ using:

$$d(i, C_k) = \|f_i - f_{C_k}\| + m\|x_i - x_{C_k}\|$$

         where $x_i$ and $x_{C_k}$ are the spatial coordinates, $f_i$ and $f_{C_k}$ are the physics-guided features.
26:         Assign mesh node $i$ to the nearest segment centroid $C_k$ if $d(i, C_k) \leq S$
27:     **end for**
28: **until** convergence or a maximum number of iterations is reached

---

Table 3: Comparison of mesh quality results with state-of-the-art methods across two datasets, where each model is independently trained for each dataset. Our approach consistently outperforms other methods across multiple aspects of predicted mesh quality, including maximum deviation, element uniformity, shape regularity, and average geometric accuracy.

| Dataset | Model | $\text{GF}_h$ $(\times 10^{-3})\downarrow$ | $\text{GF}_c$ $(\times 10^{-6})\downarrow$ | MC $(\times 10^{-3})\downarrow$ | Aspect Ratio $(\times 10^{-3})\downarrow$ |
|---|---|---|---|---|---|
| Deforming Plate | GCN | $24.0 \pm 0.6$ | $323 \pm 4$ | $11.0 \pm 0.3$ | $9.33 \pm 0.57$ |
| | $g$-U-Net | $36.1 \pm 8.5$ | $452 \pm 125$ | $20.1 \pm 0.5$ | $12.4 \pm 4.3$ |
| | MGN | $12.7 \pm 0.9$ | $248 \pm 12$ | $9.25 \pm 0.39$ | $5.34 \pm 0.26$ |
| | BSMS-GNN | $23.8 \pm 2.6$ | $170 \pm 13$ | $18.3 \pm 4.4$ | $15.4 \pm 5.9$ |
| | EAGLE | $6.75 \pm 0.8$ | $41.1 \pm 2.6$ | $5.56 \pm 0.12$ | $3.31 \pm 0.04$ |
| | MMSGN (Ours) | $\mathbf{4.43 \pm 0.08}$ | $\mathbf{7.05 \pm 1.05}$ | $\mathbf{4.78 \pm 0.09}$ | $\mathbf{2.65 \pm 0.04}$ |
| Deforming Beam | GCN | $4.91 \pm 0.36$ | $3.53 \pm 0.51$ | $54.8 \pm 8.2$ | $69.5 \pm 3.8$ |
| | $g$-U-Net | $4.91 \pm 0.50$ | $3.55 \pm 0.73$ | $34.7 \pm 1.8$ | $31.5 \pm 1.2$ |
| | MGN | $0.82 \pm 0.04$ | $0.12 \pm 0.01$ | $16.9 \pm 0.1$ | $7.43 \pm 0.10$ |
| | BSMS-GNN | $0.99 \pm 0.03$ | $0.21 \pm 0.04$ | $32.5 \pm 0.5$ | $16.1 \pm 0.3$ |
| | EAGLE | $0.64 \pm 0.04$ | $0.17 \pm 0.01$ | $5.98 \pm 0.43$ | $5.17 \pm 0.37$ |
| | MMSGN (Ours) | $\mathbf{0.33 \pm 0.01}$ | $\mathbf{0.05 \pm 0.00}$ | $\mathbf{5.24 \pm 0.04}$ | $\mathbf{3.17 \pm 0.02}$ |

DeformingPlate and 0.5 for DeformingBeam. The average cluster size $S$ is set to be $\sqrt{0.656/K}$, $\sqrt{0.125/K}$ and $\sqrt{0.005/K}$, respectively.

# D  Additional Evaluation Metrics and Results

## D.1  Additional metrics for mesh quality measure

To further enhance the evaluation of predicted mesh quality, we incorporate two additional metrics: Chamfer Distance (Wu et al., 2021) and Aspect Ratio (Zienkiewicz & Taylor, 2005). While geometric fidelity (GF) in Eq (10) measures the maximum deviation between the predicted and true meshes using the Hausdorff Distance, and mesh continuity (MC) evaluates the uniformity of mesh cell sizes, they may not fully capture the element-wise quality and average geometric discrepancies important in finite element analysis and visual computing applications. The addition of Chamfer Distance and Aspect Ratio addresses these aspects. Result comparison of state-the-art methods over all mesh quality measures are shown in Table 3.

**Chamfer Distance** – The Chamfer Distance measures the average distance between points on the predicted mesh and the true mesh, providing a balanced assessment of GF. Unlike the Hausdorff Distance, which focuses on the maximum deviation, the Chamfer Distance is sensitive to the overall distribution of errors across the mesh surfaces. As both Chamfer and Hausdorff distance are measures for GF, we name them as $\text{GF}_c$ and $\text{GF}_h$ for simplicity, respectively. The Chamfer distance is mathematically defined as:

$$\text{GF}_c(\mathcal{V}, \hat{\mathcal{V}}) = \frac{1}{|\mathcal{V}|}\sum_{\mathbf{x} \in \mathcal{V}} \min_{\hat{\mathbf{x}} \in \hat{\mathcal{V}}} \|\mathbf{x} - \hat{\mathbf{x}}\|^2 + \frac{1}{|\hat{\mathcal{V}}|}\sum_{\hat{\mathbf{x}} \in \hat{\mathcal{V}}} \min_{\mathbf{x} \in \mathcal{V}} \|\hat{\mathbf{x}} - \mathbf{x}\|^2, \tag{12}$$

where $\mathcal{V}$ and $\hat{\mathcal{V}}$ are the set of vertices in the ground-truth and predict mesh, respectively. $|\mathcal{V}|$ and $|\hat{\mathcal{V}}|$ denote the number of vertices in each mesh.

**Aspect Ratio (error)** – The Aspect Ratio metric assesses the shape quality of individual 2D or 3D mesh elements and is widely used in finite element method (FEM) literature to evaluate how closely each element approaches the ideal shape, such as an equilateral triangle or a regular tetrahedron. For example, for triangular meshes, the aspect ratio is defined as $\frac{L_{\max}}{2\sqrt{\sqrt{3}A}}$, where $L_{\max}$ is the longest edge length, $A$ is the area of the triangle. For tetrahedra mesh, it is defined as $\frac{\sqrt{6}L_{\max}}{V^{1/3}}$, where $V$ the volume of the tetrahedron. High aspect ratios indicate elongated or distorted elements, which can cause numerical instability and reduce simulation accuracy. By analyzing the aspect ratios across all elements, we can assess the overall uniformity and regularity of the mesh. To evaluate the accuracy of the predicted mesh compared to the ground truth, we calculate the aspect ratio for both the predicted and actual meshes. The Aspect Ratio Error is then determined as the $L_1$ distance between these two values. This error metric quantifies the deviation in shape quality between the predicted and

true meshes, providing a direct measure of how well the prediction preserves the ideal element shapes. Incorporating the Aspect Ratio Error allows for a more precise evaluation of mesh quality and prediction accuracy, ensuring that the segmented meshes maintain the necessary geometric properties for reliable simulations.

## D.2 Segmentation Quality Metrics

In order to rigorously evaluate the quality of our physics-informed mesh segmentation and its impact on the prediction of system dynamics, it is essential to consider metrics that assess both inter-segment and intra-segment characteristics. We introduce three such metrics —*Conductance*, *Edge Cut Ratio*, and *Silhouette Score* — which provide a comprehensive assessment of segmentation quality by quantifying the cohesion within segments and the separation between segments. The necessity of these metrics arises from the need to ensure that segments are well-separated, minimizing unnecessary interactions between dissimilar regions (inter-segment quality), and that nodes within the same segment share similar properties or behaviors (intra-segment quality).

Moreover, in our hierarchical model architecture, the intra-segment quality pertains to the micro-level information exchange stage. High intra-segment quality facilitates accurate modeling of local dynamics within each segment by ensuring that nodes are cohesive and share similar dynamic behaviors. Conversely, the inter-segment quality directly relates to the macro-level information exchange stage. High inter-segment quality ensures efficient communication between segments by reducing redundant or irrelevant interactions, which is crucial for capturing global dynamics across the entire mesh. Below are the details of three metrics to measure segmentation quality.

**Conductance** – Conductance measures the fraction of total edge connections that cross between different segments relative to the total connections of the segments. It assesses how well the segmentation minimizes inter-segment connections while maintaining intra-segment cohesion. Let $G = (\mathcal{V}, \mathcal{E})$ as an undirected graph representing the mesh, where $\mathcal{V}$ is the set of nodes and $\mathcal{E}$ is the set of edges. Let $S$ be a segment and $\bar{S} = G \setminus S$ be its complement. The conductance of segment $S$ is defined as:

$$\text{Conductance} = \frac{\left|\{(u,v) \in \mathcal{E} \mid u \in S, \ v \in \bar{S}\}\right|}{\min\left(\text{vol}(S), \ \text{vol}(\bar{S})\right)}, \tag{13}$$

where the numerator is the number of edges crossing between $S$ and $\bar{S}$. The volumn of segment $S$ is given by $\text{vol}(S) = \sum_{u \in S} \deg(u)$, where $\deg u$ is the degree of node $u$ (the number of edges connected to $u$).

**Edge Cut Ratio** – The Edge Cut Ratio quantifies the proportion of edges that are cut by the segmentation relative to the total number of edges in the mesh. It is defined as:

$$\text{Edge Cut Ratio} = \frac{|\{(u,v) \in \mathcal{E} \mid \text{Seg}(u) \neq \text{Seg}(v)\}|}{E}, \tag{14}$$

where the denominator is the number of edges that connect nodes in different segment. $\text{Seg}(u)$ denotes the segment to which node $u$ belongs and $E = |\mathcal{E}|$ is the total number of edges.

**Silhouette Score** – For each node $i$, the Silhouette Score evaluates how similar $i$ is to nodes in its own segment compared to nodes in other segments. It is defined as:

$$\text{Silhouette Score} = \frac{1}{N} \sum_{u=1}^{N} \frac{b(i) - a(i)}{\max\{a(i), \ b(i)\}}, \tag{15}$$

where $N$ is the total number of nodes, $a(i)$ is the average dissimilarity of node $i$ with all other nodes in the same segment and $b(i)$ is the lowest average dissimilarity of node $i$ to any other segment to which $i$ does not belong. To be more specific $a(i) = \frac{1}{|S_i|-1} \sum_{\substack{j \in S_i \\ j \neq i}} d(i,j)$, $b(i) = \min_{S' \neq S_i} \left( \frac{1}{|S'|} \sum_{j \in S'} d(i,j) \right)$, where $S_i$ is the segment containing node $i$ and $d(i,j)$ can be any appropriate distance metric, such as Euclidean distance based on node features or positions.

By combining these metrics, we achieve a comprehensive evaluation of segmentation quality that covers both the internal cohesion of segments and their external separation. Having these metrics, along with prediction result metrics, can better help us understand the effect of segmentation on

the predicted system dynamics. These metrics can be used to help finding better physics-informed segment features and determining the optimal segmentation number (results and discussion in Appendix E.2).

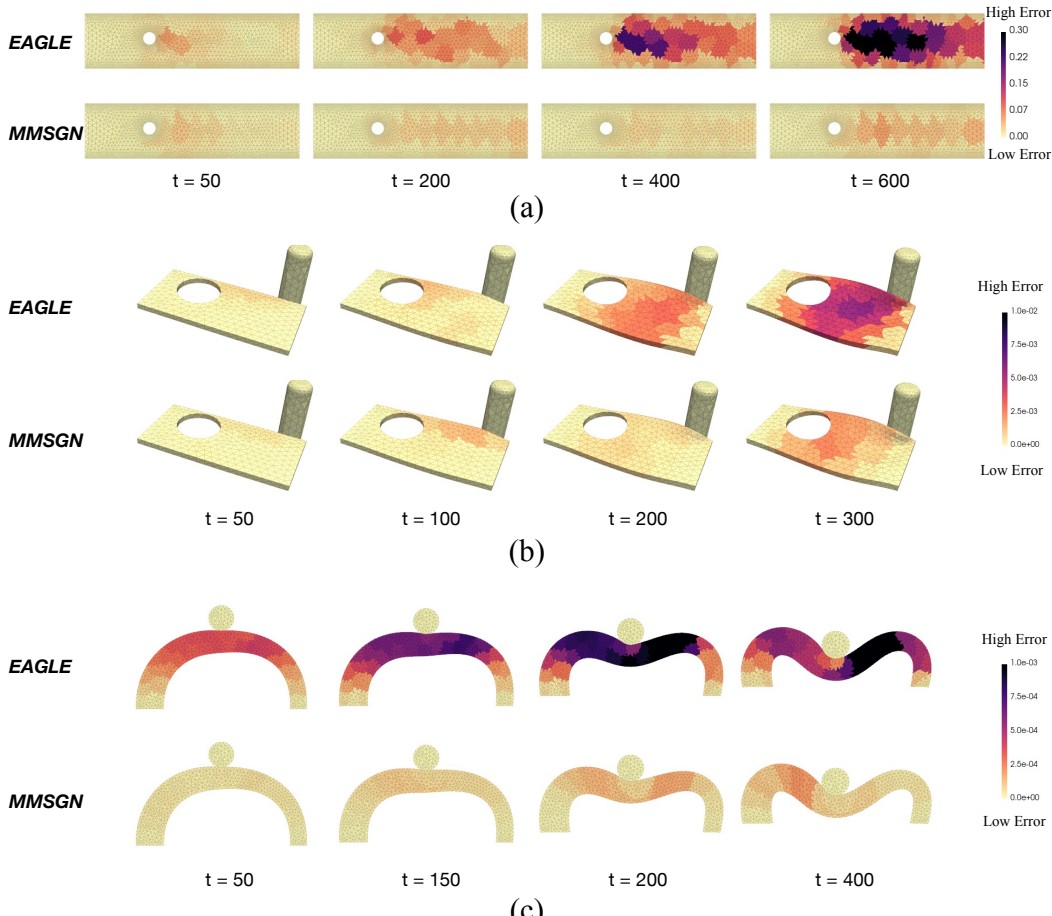

Figure 6: Visualization of simulation rollouts over time for three datasets, comparing our segmentation method with EAGLE. Nodes are colored based on the average prediction error within their segments. Our method consistently produces uniform segment colors across time steps, indicating that nodes within each segment share similar dynamic behaviors and that segments maintain high continuity. For example, in (a), segmentation follows periodic wave patterns in fluid dynamics, while in (c), it reflects symmetrical system dynamics with symmetric segment coloring. These visualizations demonstrate that our segmentation effectively captures the temporal and spatial dynamics of the system, outperforming EAGLE.

### D.3 VISUALIZATION OF SEGMENTATION ALIGNMENT WITH SYSTEM DYNAMICS

To visualize the how predict mesh properties in each segment various through time for different method, we include additional visualization results in Figure 6, where mesh nodes at each predicted time step are colored based on the average prediction error within their segments. In all three datasets, our method produces segments with uniform colors across time steps, indicating that nodes within the same segment share similar dynamic behaviors and different segments have little discrepancies or maintain high continuity. This means that our segmentation effectively groups regions with coherent dynamic interactions, ensuring consistent modeling and accurate prediction of the system's evolution over time. Such consistent segmentation enhances the model's ability to capture and represent the underlying physical properties, leading to more reliable and stable simulation outcomes. Specifically, our methods is able to accurately capture dynamic patterns of periodic wake formations shown in

Figure 6(a). Also, in case with inherent symmetry, such as the Figure 6(c), our method successfully generates symmetric segments that share similar dynamic properties.

Our segmentation method effectively aligns with system dynamics but could be sensitive to initial setup parameters, potentially missing important physical interdependencies in complex systems. To address this, future work should incorporate modal analysis and adaptive clustering techniques. These enhancements would improve the physical relevance and robustness of our segmentation, making it more versatile for a wider range of dynamic environments.

# E  ABLATION STUDIES

## E.1  MICRO-LEVEL INFORMATION EXCHANGE

According to Figure 7, with fewer message passing steps, each node updates only based on immediate neighbors, resulting in higher prediction errors and mesh discontinuities. As more steps are introduced, nodes gather information from a broader neighborhood, leading to more accurate predictions and smoother mesh transitions. The early iterations of message passing yield the most noticeable improvements, as nodes rapidly gather useful information from their surrounding environment. Later iterations primarily serve to fine-tune the mesh continuity and reduce local errors, but the impact on overall accuracy diminishes. Interestingly, increasing the number of message-passing steps beyond a certain point continues to improve mesh quality, but prediction accuracy may degrade. This suggests the occurrence of oversmoothing, where the model excessively homogenizes node features, or overfitting, where the model starts to memorize local information rather than generalize. This phenomenon highlights the importance of carefully selecting the number of message-passing steps during micro-level information exchange step to strike the right balance between improving prediction accuracy and maintaining mesh quality.

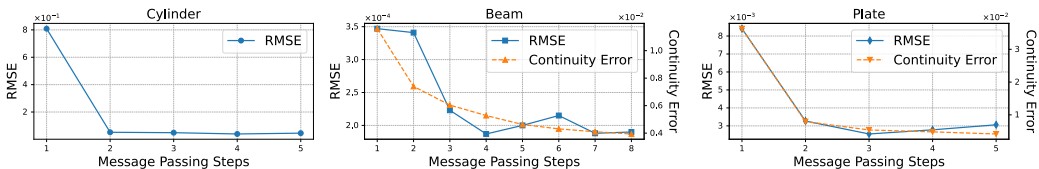

Figure 7: Ablation study on the impact of varying message-passing steps in the micro-level information exchange on prediction performance across three datasets.

## E.2  MACRO-LEVEL INFORMATION EXCHANGE

**Comparison of Different Segment Extraction Methods** – As shown in Table 4, mesh segment method can have a large impact on the result. By comparing SLIC and METIS results, we find a 28%, 21% and 14% improvement in RMSE-all for CylinderFLow, DeformingPlate and Deforming-Beam. The best segment method for CylinderFlow is SLIC-OD-EXP. The best segment method for DeformingPlate and DeformingBeam is SLIC-ODBD-EXP.

To thoroughly evaluate the different segmentation methods, we utilize the three metrics -Conductance, Edge Cut Ratio, and Silhouette Score - introduced in Appendix D.2 to assess both inter-segment and intra-segment qualities of mesh partitions, providing a comprehensive understanding of each method's effectiveness. We then analyzed the correlation between these segmentation metrics and overall dynamic system performance, including mesh quality and prediction error, as illustrated in Figure 9 (a-c). Our findings indicate that segmentation methods incorporating physics-informed features, particularly those utilizing both obstacle and boundary distances with exponential transformations, generally enhance model performance across various datasets. This improvement can be attributed to three key factors: (1) *Alignment with Dynamics*, where segmentation reflecting physical influences enables more effective learning of the system's dynamics; (2) *Enhanced Segment Quality*, achieved through improved intra-segment cohesion and minimized inter-segment interactions, facilitating better learning of localized patterns; and (3) *Benefit to Learning*, where emphasizing critical regions via exponential transformations allows the model to focus on areas with significant dynamic changes,

Table 4: Ablation study on different segment extraction methods over different dataset.

| Segmentation Method | Dataset | GF$_h$ ↓ | GF$_c$ ↓ | MC ↓ | Aspect Ratio ↓ | RMSE-1 | RMSE-all |
|---|---|---|---|---|---|---|---|
| METIS | Cylinder | - | - | - | - | 3.44e-03 | 4.59e-02 |
| | Plate | 5.32e-03 | 1.36e-05 | 5.33e-03 | 2.97e-03 | 2.67e-04 | 3.29e-03 |
| | Beam | 3.88e-04 | 5.61e-08 | 5.18e-03 | 3.09e-03 | 1.15e-05 | 2.16e-04 |
| SLIC-OD-log | Cylinder | - | - | - | - | 3.31e-03 | 4.16e-02 |
| | Plate | 4.53e-03 | 7.28e-06 | 5.02e-03 | 2.75e-03 | 2.72e-04 | 2.87e-03 |
| | Beam | 3.95e-04 | 5.44e-08 | 5.33e-03 | 3.30e-03 | 1.18e-05 | 2.68e-04 |
| SLIC-OD-exp | Cylinder | - | - | - | - | 2.88e-03 | 3.34e-02 |
| | Plate | 5.38e-03 | 1.35e-05 | 4.62e-03 | 2.62e-03 | 2.67e-04 | 3.62e-03 |
| | Beam | 3.81e-04 | 5.68e-08 | 5.40e-03 | 3.32e-03 | 1.20e-05 | 2.51e-04 |
| SLIC-ODBD-log | Cylinder | - | - | - | - | 2.99e-03 | 5.57e-02 |
| | Plate | 5.11e-03 | 1.26e-05 | 4.93e-03 | 2.76e-03 | 2.78e-04 | 3.57e-03 |
| | Beam | 3.74e-04 | 5.48e-08 | 5.28e-03 | 3.26e-03 | 1.18e-05 | 2.46e-04 |
| SLIC-ODBD-exp | Cylinder | - | - | - | - | 2.95e-03 | 4.37e-02 |
| | Plate | 4.33e-03 | 6.85e-06 | 4.66e-03 | 2.63e-03 | 2.66e-04 | 2.57e-03 |
| | Beam | 3.21e-04 | 4.59e-08 | 5.28e-03 | 3.20e-03 | 1.17e-05 | 1.87e-04 |

thereby enhancing prediction accuracy. These results demonstrate that the choice of segmentation method impacts the model's ability to learn dynamic behaviors, and the introduction of additional metrics reveals that physics-informed segmentation effectively aligns mesh partitions with the system's inherent physical properties, thereby benefiting the learning process.

**Influence of Segment Count on Performance** – Table 5 and Table 6 present the RMSE-1, RMSE-all, and various mesh quality metrics as the total number of mesh segments is varied during training on three different datasets. In general, MMSGN maintains stable performance with relatively low variance, indicating that results are not highly sensitive to segment count. This robustness ensures reliable accuracy across different mesh granularities. However, increasing the number of segments—thereby reducing finite elements per segment—can lead to slight decreases in accuracy and performance.

To comprehensively evaluate the effect of segment number and determine the optimal segmentation for a given dataset, we analyzed prediction accuracy across a wide range of segment counts (from 3 to 51) during training on the DeformingBeam dataset. The impact of varying the number of mesh segments on prediction accuracy is illustrated in Figure 8 and Figure 9(d). According to the plots, we identify 19 segments as the optimal number. At this segmentation level, the model achieves the lowest RMSE and Chamfer Distance, indicating high prediction accuracy and precise shape representation. The Hausdorff Distance is also minimized, reflecting excellent alignment between the predicted and true meshes. While the Silhouette score peaks at 9 segments—suggesting well-defined and compact clusters—the slight decrease at 19 segments is offset by significant gains in other performance metrics. Choosing a lower number of segments, such as 3 or 9, may result in higher Silhouette scores but can compromise mesh detail and prediction accuracy due to insufficient spatial granularity. Conversely, selecting a higher number of segments beyond 19 shows diminishing returns, with only marginal improvements or slight degradations in some metrics and a continued decline in Silhouette scores, potentially indicating over-segmentation and unnecessary computational complexity.

In conclusion, when presented with a new dataset, the optimal number of segments can be determined by first computing Silhouette scores for various segment counts to assess cluster cohesion and separation without requiring model training. This provides initial guidance on meaningful segmentation levels. Subsequently, training the model with different segment numbers and evaluating performance metrics like RMSE, Hausdorff Distance, and Chamfer Distance will help identify the point where performance improvements plateau or begin to reverse, indicating the optimal balance between segmentation detail and model efficacy.

**Influence of Positional Encoding on Performance** – Table 6 and Figure 10(a) shows the effect of adding positional encoding for small and large number of segments across three dataset. According to the results, we identified several key findings. Firstly, the effectiveness of PE depends on the number of segments: in the CylinderFlow and Deforming Plate datasets, incorporating PE with fewer segments improves performance across multiple metrics by reducing positional ambiguity.

With low segment counts, each segment covers larger, more diverse areas, limiting the model's spatial detail and understanding of segment relationships. PE provides explicit positional information, allowing the model to distinguish distinct regions within the same segment and better comprehend their interactions. However, as the number of segments increases and spatial resolution improves, the benefits of PE diminish and may even introduce unnecessary complexity that hinders performance. Additionally, dataset-specific factors influence PE's effectiveness; for example, the DeformingBeam dataset, with its complex geometry and deformation, did not benefit from PE. This indicates that PE's success depends not only on segment count but also on how well the PE implementation aligns with the dataset's unique characteristics.

Consequently, tailored PE approaches that consider specific geometry and deformation patterns are necessary for complex systems to achieve significant performance gains. In summary, while PE enhances the performance of graph-based networks, further advancements are needed to develop optimal encoding strategies that consistently improve performance across diverse dynamic systems.

**Influence of Segment Overlap on Performance** – Table 5 and Figure 10(b) illustrate the effect of adding segment overlap for small and large number of segments across three datasets. According to the results, the effectiveness of adding overlap between segments ($\delta > 0$) depends on both the segment count and the characteristics of the dataset, such as dimensionality, mesh type, and system dynamics. Overlapping segments are more beneficial with higher segment counts where discontinuities are more prevalent. In Eulerian systems, overlaps enhance the capture of complex interactions and smooth transitions on fixed meshes, leading to improved representation of fluid dynamics. Conversely, in Lagrangian systems where meshes move with the material, overlaps can create redundancy and complicate connectivity, with their impact on model performance varying based on mesh structures and deformation behaviors. For example, in the Deforming Beam dataset, which uses a prism mesh suited for directional deformation, overlapping segments improve performance by facilitating smooth transitions along its mesh surface, especially with a higher number of segments. In contrast, the Deforming Plate dataset employs a tetrahedral mesh with complex, isotropic deformations, where overlaps introduce unnecessary complexity and redundancy, resulting in decreased performance. Therefore, despite both being 3D Lagrangian systems, the different mesh types and deformation patterns explain why overlapping segments benefit the Deforming Beam but not the Deforming Plate.

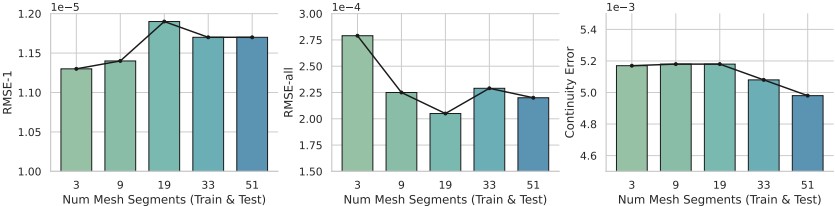

Figure 8: Impact of varying mesh segment numbers during training on prediction accuracy under the DeformingBeam dataset. The number of mesh segments remains consistent during both training and testing. In general, MMSGN maintains stable performance with relatively low variance, indicating that results are not highly sensitive to segment count. This robustness ensures reliable accuracy across different mesh granularities. However, increasing the number of segments—thereby reducing finite elements per segment—can lead to slight decreases in accuracy and performance. More detailed analysis on the effect of segmentation numbers to various metrics can be found in Figure 9(d).

# F GENERALIZATION STUDIES

To evaluate the generalizability of our MMSGN model, we created a larger-scale DeformingBeam dataset, detailed in Appendix A.

## F.1 PERFORMANCE ON LARGER-SCALE DATASETS

Table 7 summarizes the generalization performance of various models trained on the DeformingBeam dataset and directly applied to DeformingBeam(large), a scaled-up version. The results demonstrate that MMSGN consistently outperforms all other models across all metrics. In terms of mesh quality, MMSGN achieves a 51% improvement over EAGLE and a 52% improvement over BSMS

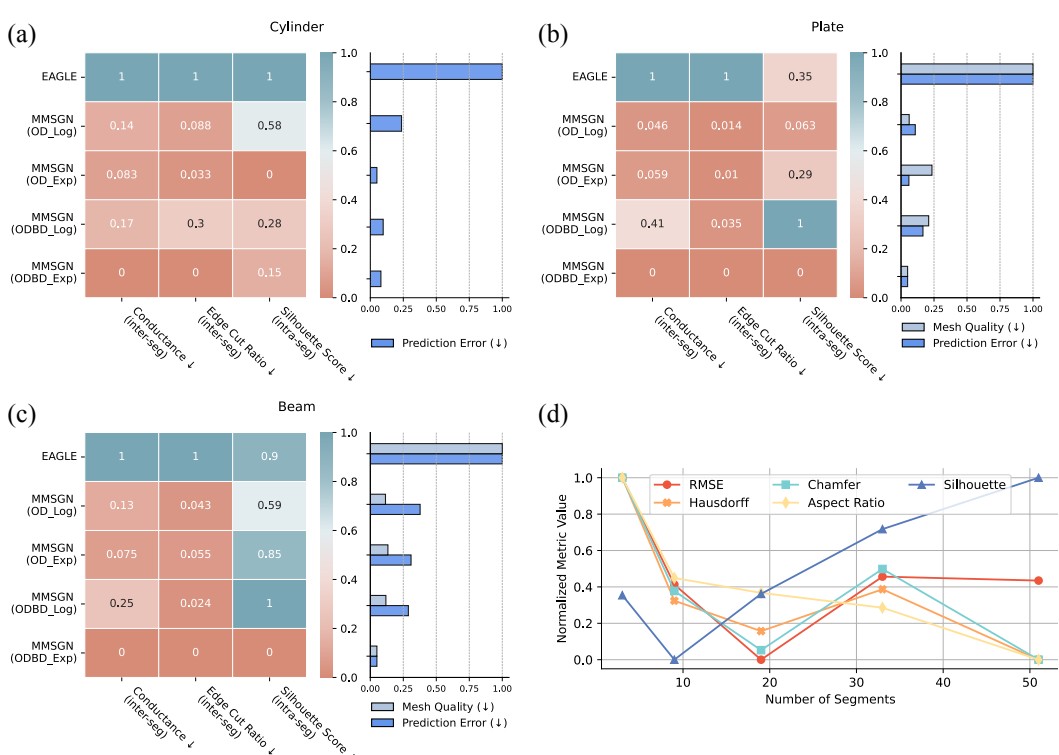

Figure 9: **(a-c)** Evaluation of different segmentation methods under three datasets. The heatmap (left) presents normalized Conductance, Edge Cut Ratio, and reversed Silhouette Score for EAGLE and four MMSGN variants. Metrics are scaled between 0 and 1, with Silhouette Scores reversed to ensure consistent evaluation criteria, where lower values indicate better segmentation quality. The sidebar plot (right) depicts normalized Prediction Error and Mesh Quality, with a minimum value of 0.05 applied to avoid invisible bars. These figures evaluate segmentation quality across multiple metrics and demonstrate how different segmentation methods influence model accuracy and mesh quality, emphasizing the advantages of our physics-informed segmentation strategies; **(d)** Dependence of various performance metrics on the number of segments in MMSGN under Deforming Beam dataset. The plot illustrates how the normalized values of several performance metrics vary with the number of segments. Each metric is represented by a distinct curve, demonstrating the relationship between segment number and overall performance. This figure evaluates the effect of segment number and guides the selection of the optimal number of segments for balanced performance across all metrics.

Table 5: Ablation study of number of segments, and effect of adding segment overlap.

| DATASET | $N_{\text{SEG}}$ | $\delta > 0$ | $GF_h \downarrow$ | $GF_c \downarrow$ | MC ↓ | ASPECT RATIO ↓ | RMSE-1 | RMSE-ALL |
|---|---|---|---|---|---|---|---|---|
| CYLINDER | 16 | ✗ | - | - | - | - | 3.07E-03 | 3.96E-02 |
|  | 16 | ✓ | - | - | - | - | 3.02E-03 | 4.09E-02 |
|  | 36 | ✗ | - | - | - | - | 3.58E-03 | 4.63E-02 |
|  | 36 | ✓ | - | - | - | - | 2.88E-03 | 3.34E-02 |
| PLATE | 9 | ✗ | 4.89E-03 | 9.56E-06 | 5.01E-03 | 2.78E-03 | 2.83E-04 | 3.05E-03 |
|  | 9 | ✓ | 5.96E-03 | 1.34E-05 | 5.10E-03 | 2.90E-03 | 2.75E-04 | 3.64E-03 |
|  | 19 | ✗ | 4.33E-03 | 6.85E-06 | 4.66E-03 | 2.63E-03 | 2.66E-04 | 2.57E-03 |
|  | 19 | ✓ | 5.08E-03 | 2.18E-05 | 4.76E-03 | 2.72E-03 | 2.65E-04 | 4.49E-03 |
| BEAM | 9 | ✗ | 3.45E-04 | 5.21E-08 | 5.18E-03 | 3.30E-03 | 1.14E-05 | 2.25E-04 |
|  | 9 | ✓ | 3.67E-04 | 5.38E-08 | 5.25E-03 | 3.32E-03 | 1.18E-05 | 2.37E-04 |
|  | 19 | ✗ | 3.29E-04 | 4.81E-08 | 5.18E-03 | 3.26E-03 | 1.19E-05 | 2.05E-04 |
|  | 19 | ✓ | 3.21E-04 | 4.59E-08 | 5.28E-03 | 3.20E-03 | 1.17E-05 | 1.87E-04 |

Table 6: Ablation study of number of segments and whether to add PE or not.

| DATASET | $N_{\text{SEG}}$ | PE | $GF_h \downarrow$ | $GF_c \downarrow$ | MC ↓ | ASPECT RATIO ↓ | RMSE-1 | RMSE-ALL |
|---|---|---|---|---|---|---|---|---|
| CYLINDER | 16 | ✗ | - | - | - | - | 3.02E-03 | 4.09E-02 |
|  | 16 | ✓ | - | - | - | - | 2.95E-03 | 3.69E-02 |
|  | 36 | ✗ | - | - | - | - | 2.88E-03 | 3.34E-02 |
|  | 36 | ✓ | - | - | - | - | 3.23E-03 | 4.10E-02 |
| PLATE | 9 | ✗ | 4.89E-03 | 9.56E-06 | 5.01E-03 | 2.78E-03 | 2.83E-04 | 3.05E-03 |
|  | 9 | ✓ | 4.39E-03 | 6.94E-06 | 5.11E-03 | 2.98E-03 | 2.86E-04 | 2.57E-03 |
|  | 19 | ✗ | 4.33E-03 | 6.85E-06 | 4.66E-03 | 2.63E-03 | 2.66E-04 | 2.57E-03 |
|  | 19 | ✓ | 5.21E-03 | 1.34E-05 | 4.76E-03 | 2.62E-03 | 2.81E-04 | 3.67E-03 |
| BEAM | 9 | ✗ | 3.67E-04 | 5.38E-08 | 5.25E-03 | 3.32E-03 | 1.18E-05 | 2.37E-04 |
|  | 9 | ✓ | 3.93E-04 | 6.54E-08 | 5.32E-03 | 3.43E-03 | 1.15E-05 | 2.58E-04 |
|  | 19 | ✗ | 3.21E-04 | 4.59E-08 | 5.28E-03 | 3.20E-03 | 1.17E-05 | 1.87E-04 |
|  | 19 | ✓ | 3.33E-04 | 5.15E-08 | 5.22E-03 | 3.27E-03 | 1.17E-05 | 1.88E-04 |

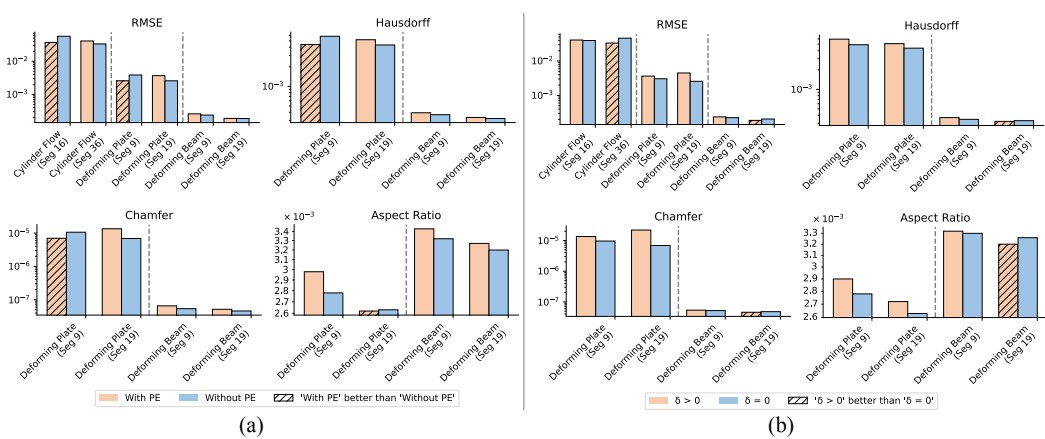

(a)    (b)

Figure 10: Ablation study on the effects of position encoding and segment overlap across datasets with varying segment numbers. The figure presents the performance metrics for models both with and without the position encoder (a), and with and without considering segment overlap (b) across three distinct datasets, each characterized by a different number of segments. By comparing these conditions, the study highlights how the inclusion of position encoding and the handling of segment overlap influence overall performance, thereby informing the selection of optimal model configurations.

for Geometric Fidelity (GF). Similarly, for Mesh Continuity (MC), MMSGN achieves the best performance with a value of 1.13e-02, representing a 43% improvement over EAGLE, the next-best model. For the RMSE metrics, MMSGN delivers the lowest RMSE-1, RMSE-50, and RMSE-all. Notably, MMSGN's RMSE-all is 45% lower than both MGN and EAGLE. These findings suggest that MMSGN not only preserves prediction accuracy but also enhances mesh quality when generalizing to larger-scale data, significantly surpassing state-of-the-art models in both accuracy and mesh quality. This demonstrates MMSGN's robust generalization ability, making it highly suitable for complex, large-scale dynamic systems.

Table 7: Generalization performance of our method and five baseline models on the scaled-up DeformingBeam dataset. MMSGN demonstrates superior accuracy and mesh quality when generalizing to an unseen dataset with a denser mesh and more extensive long-range dynamic effects.

| METHOD | $GF_h \downarrow$ | $GF_c \downarrow$ | MC $\downarrow$ | ASPECT RATIO $\downarrow$ | RMSE-1 | RMSE-50 | RMSE-ALL |
|---|---|---|---|---|---|---|---|
| GCN | 2.18E-02 | 3.28E-05 | 1.21E-01 | 1.69E-01 | 2.57E-04 | 1.95E-03 | 1.11E-02 |
| $g$-U-NET | 1.94E-02 | 2.80E-05 | 4.56E-02 | 7.01E-02 | 1.60E-04 | 1.87E-03 | 1.01E-02 |
| MGN | 2.32E-02 | 1.43E-05 | 2.00E-02 | 2.57E-02 | 1.34E-04 | 1.43E-03 | 6.42E-03 |
| BSMS | 1.72E-02 | 3.34E-05 | 1.35E-01 | 1.17E-01 | 4.47E-04 | 3.19E-03 | 1.03E-02 |
| EAGLE | 1.69E-02 | 2.20E-05 | 1.98E-02 | 5.15E-02 | 8.42E-05 | 1.45E-03 | 8.37E-03 |
| MMSGN | **8.25E-03** | **5.59E-06** | **1.13E-02** | **2.16E-02** | **5.84E-05** | **9.43E-04** | **4.59E-03** |

### F.2 EFFECT OF MESH SEGMENT COUNT ON GENERALIZATION

**Generalization with Varying Segment Counts During Testing** – Across three datasets, we perform generalization studies where the model is tested using a varying number of segments. The results in Figure 11 illustrate the generalization performance. Pink columns are the references for regular testing and the others are generalization to different number of segments from training. Overall, the MMSGN model can generalize very well to different number of segments during testing.

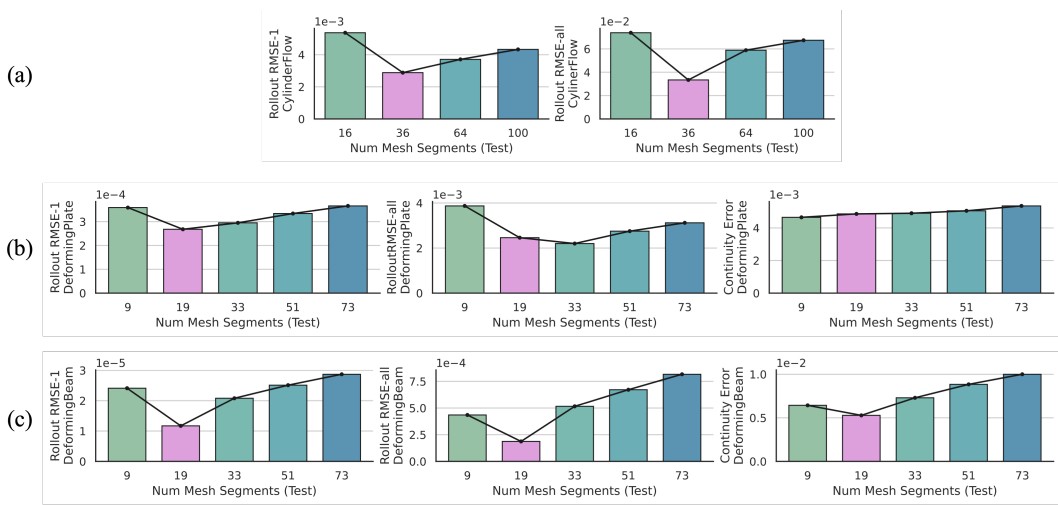

Figure 11: Generalization performance of our method under varying segment counts during testing over three datasets. (a) CylinderFlow: effect of number of segments for test set on different metrics, where model is trained under 36 segments (colored in pink); (b) DeformingPlate: effect of number of segments for test set on different metrics, where model is trained under 19 segments (colored in pink); (c) DeformingBeam: effect of number of segments for test set on different metrics, where model is trained under 19 segments (colored in pink). This figure illustrates that our MMSGN model, despite being trained with a fixed number of mesh segments, maintains strong accuracy and mesh quality when tested with varying numbers of mesh segments.

**Impact of Segment Count During Training and Testing** – Equipped with message passing and transformer mechanisms, MMSGN can handle an arbitrary number of segments. Figure 12 shows

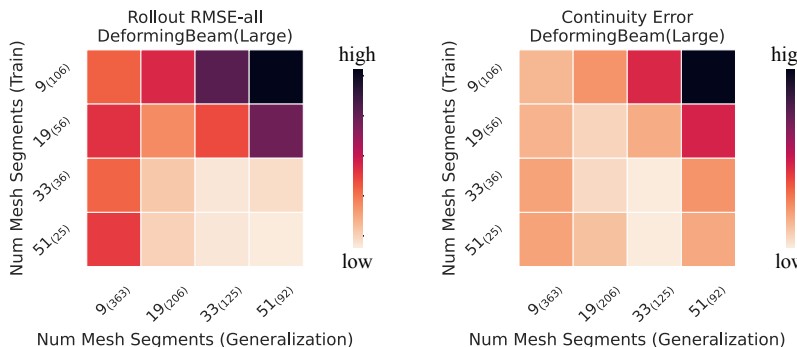

Figure 12: Generalization performance of our method on larger domains under different number of mesh segmentation during training and testing. The subscript of each mesh segment indicating the average number of nodes per segment. MMSGN demonstrates robustness and adaptability in handling larger domains with varying mesh segments, making it well-suited for real-world applications involving large and complex mesh structures.

Table 8: Comprehensive evaluation of our method alongside MGN, BSMS, and EAGLE under three datasets. MMSGN consistently delivers stable, competitive efficiency while maintaining high accuracy and superior mesh quality.

| Dataset | Model | RMSE-all | MC ↓ | Train Time per step [ms] ↓ | Train Memory [MB] ↓ | Test Time per step [ms] ↓ | Test Memory [MB] ↓ | Train Time total [h] ↓ |
|---|---|---|---|---|---|---|---|---|
| Cylinder | MGN | 4.81e-02 | - | 66.7 | 698.5 | 20.2 | 67.2 | 37.1 |
| | BSMS | 1.37e-01 | - | 54.7 | 430.3 | 23.8 | 57.9 | 30.4 |
| | EAGLE | 5.83e-02 | - | 69.5 | 618.7 | 28.8 | 230.8 | 38.6 |
| | MMSGN | 3.72e-02 | - | 56.2 | 366.6 | 20.0 | 65.0 | 31.2 |
| Plate | MGN | 1.47e-02 | 9.25e-03 | 131.9 | 6021.5 | 36.2 | 445.5 | 73.3 |
| | BSMS | 1.18e-02 | 1.83e-02 | 83.9 | 910.1 | 37.7 | 77.9 | 46.6 |
| | EAGLE | 3.87e-03 | 5.56e-03 | 81.2 | 1090.8 | 32.4 | 362.7 | 45.1 |
| | MMSGN | 2.60e-03 | 4.78e-03 | 76.5 | 648.1 | 29.3 | 103.3 | 42.5 |
| Beam | MGN | 4.72e-04 | 1.69e-02 | 79.1 | 1074.4 | 28.6 | 83.8 | 22.0 |
| | BSMS | 4.98e-04 | 3.25e-02 | 61.8 | 213.7 | 30.7 | 35.6 | 17.2 |
| | EAGLE | 4.22e-04 | 5.98e-03 | 53.5 | 410.3 | 26.0 | 153.5 | 14.9 |
| | MMSGN | 2.07e-04 | 5.24e-03 | 53.4 | 234.5 | 24.2 | 47.1 | 14.8 |

the generalization performance of our MMSGN model to larger domain as heatmaps, where models trained with a specific number of segment under deformingBeam dataset are tested with varying number of segments under deformingBeam (large). We observe that better results are seen when the number of nodes per segment during training is less than or equal to that in the generalizing domain, or when the number of segments is greater. Overall, we demonstrate MMSGN's robustness and adaptability in generalizing to larger domains with varying mesh segments, making it highly suitable for real-world applications involving large and diverse mesh graphs.

# G COMPUTATIONAL EFFICIENCY ANALYSIS

Table 8 listed the training time, test time and number of parameters for four models MGN, BSMS-GNN, EAGLE and MMSGN across three datasets. The RMSE-all is also listed as performance reference. Our MMSGN model has comparable or better efficiency compared with other models. Notably, the MMSGN model has exceptional efficiency with RMSE-all better than other baselines.

# H QUALITATIVE RESULTS

Figure 13, 14, 15, and 16 illustrate selected rollout results for all three datasets under different models.

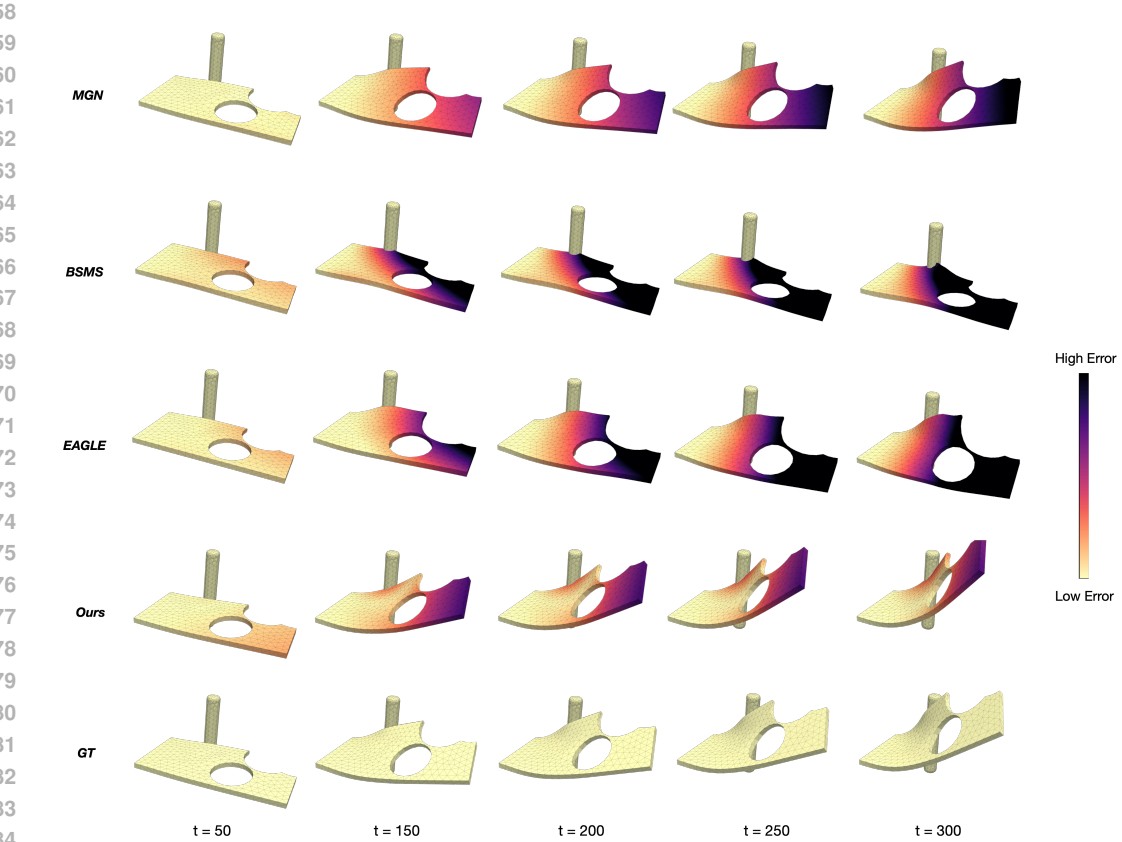

Figure 13: Additional simulation results for different models under DeformingPlate dataset.

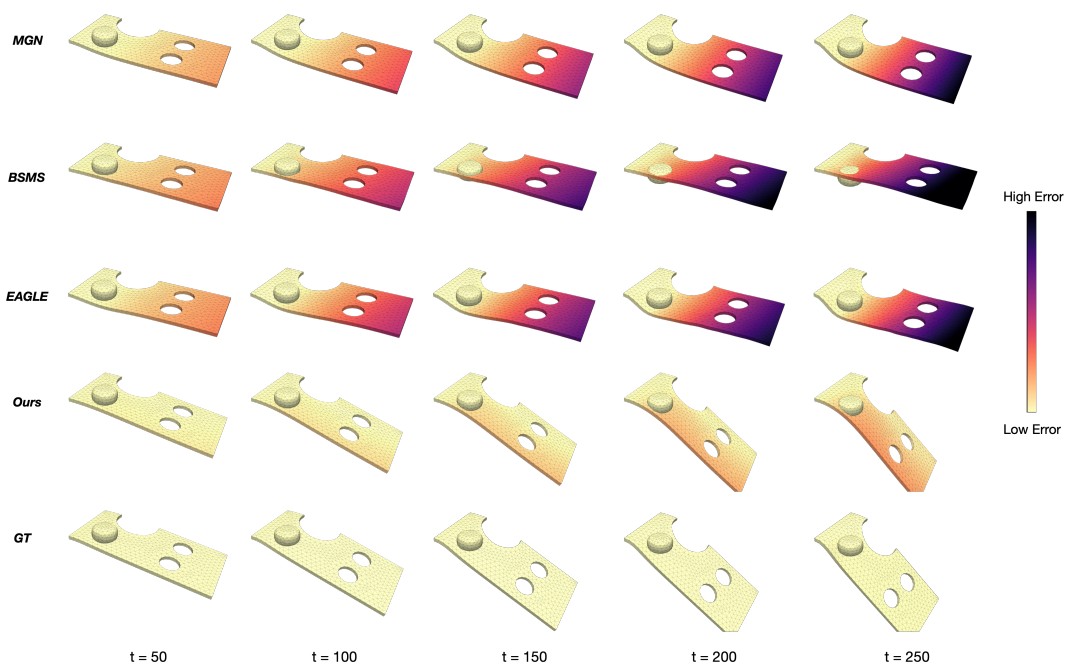

Figure 14: Additional simulation results for different models under DeformingPlate dataset.

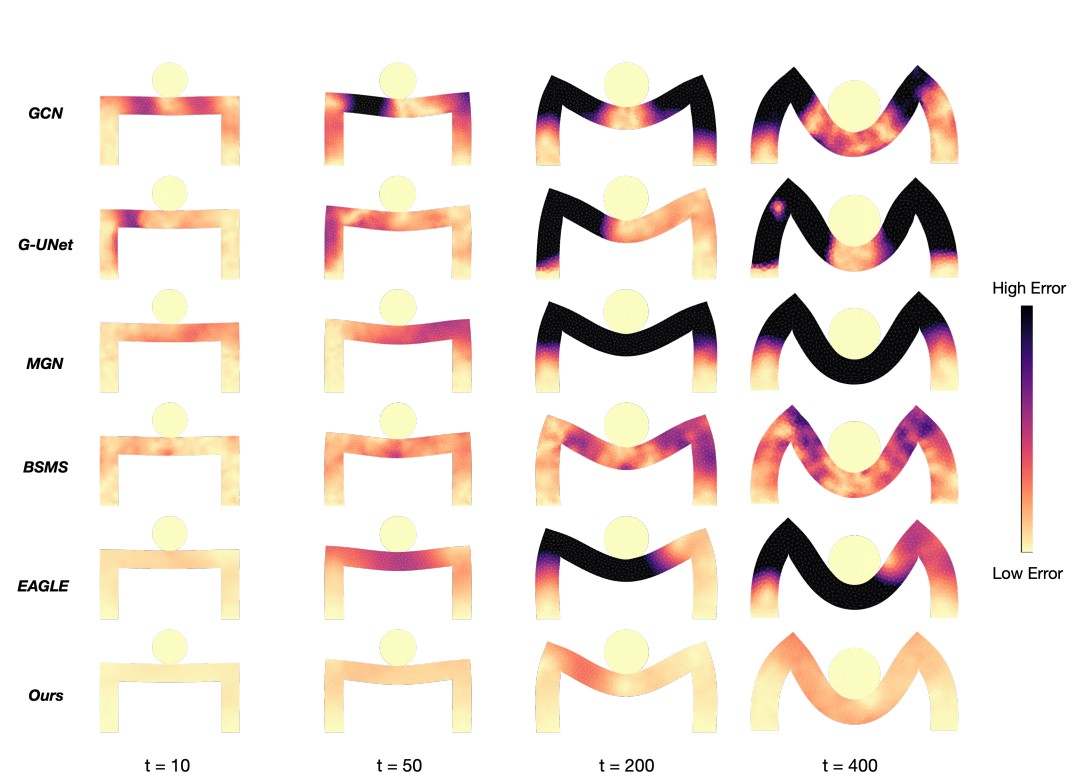

Figure 15: Additional simulation results for different models under DeformingBeam dataset

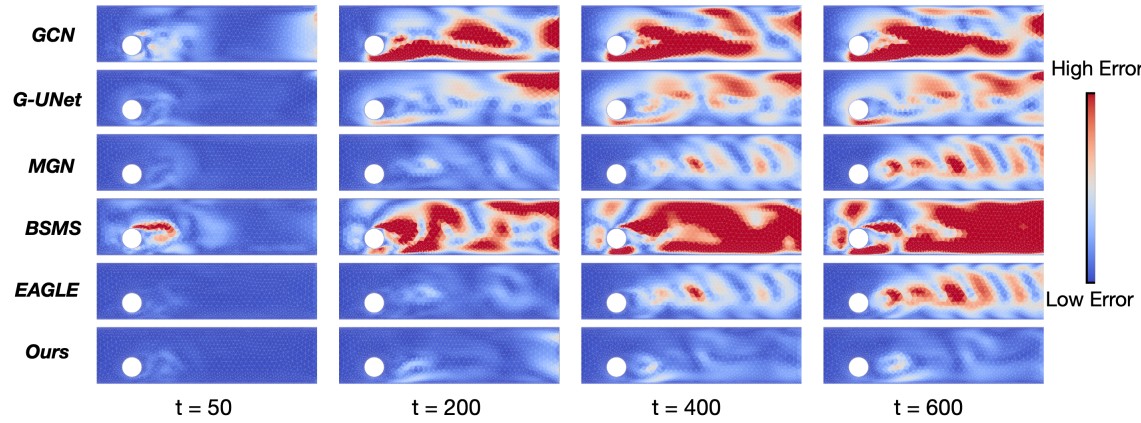

Figure 16: Additional simulation results for different models under CylinderFlow dataset

