# OpenReview forum: "Physically Aligned Hierarchical Mesh-based Network for Dynamic System Simulation"
_ICLR.cc/2025/Conference — Submitted to ICLR 2025_

### Official Review · Reviewer_YvcY · 2024-10-31

**Soundness:** 3
**Presentation:** 3
**Contribution:** 1
**Rating:** 5
**Confidence:** 4

**Summary:**

The paper builds upon the EAGLE model and experiments different improved clustering techniques against three datasets for physics simulation on irregular meshes. The results show that some modifications perform better on some task, but not on others.

**Strengths:**

The paper is mostly well written and easy to follow. The related work section and the references to the state-of-the-art are relevant and in sufficient amount. The authors conducted an important number of ablations and announced public availability of the code and dataset, which is a good initiative, especially since the paper introduces a new benchmark.

**Weaknesses:**

**Major**
My main concerns are the contributions of the paper with regards to EAGLE and the clarity of these contributions in the paper. The approach is extremely close to the EAGLE model, and while results are indeed better, the better barely explain why, and I struggle to extract general knowledge that would be transferable to future research. To be precise:
- The fact that Multi-level is highly beneficial in models for simulating physics on meshes is well known from previous results (Liu et al, 2021, Janny et al 2023, Cao et al, 2023).
- The "micro-level" structure is a straightforward MGN model, exactly like in EAGLE.
- The "macro-level" structure is an attention over a fully connected mesh on the downscaled graph, exactly like in EAGLE.
- The merge of micro and macro level information is done by concatenating both representations, like in EAGLE.

Eventually, I noted the following differences with Janny et al. 2023 :
- The clustering algorithm is improved. To me, this is the main contribution of the paper, I will come back to this.
- The position encodings seem improved, but Table 5 indicates that it does not have impact on the performances.
- Cluster representations are aggregated via average pooling, while EAGLE was using a GRU. This seems like a minor modification, and it is not discussed in the paper.
- Finally, the projection of the concatenated representation to the physical space is done with an MLP instead of another layer of GNN. Again, this difference is minor and not discussed in the paper.

Hence the only key contribution is the new clustering algorithm which uses physical prior to downscale the mesh while EAGLE was using a simple position-based clustering. This a valid and interesting way of improving this kind of model. Yet, the paper limits itself to describe when it works on three datasets, and fail (in my opinion) to extract rules, knowledge, insights that would advances the field of research.
- Algorithm 1 seems to be a straightforward K-means algorithm with handcrafted features modeling some aspect of the physical property of the domain.
- The authors tested different features, some of them working on some datasets, some did not without explanations. I could not find any insights on this phenomenon that would explain why and how the contribution actually benefits the learning of the dynamics.
- While the authors claims that the model works because the clustering "ensure that similar physical interactions are handled uniformly", I see no evidence that these clusters make indeed more sense for training an DL based simulator, apart from empirical results on three datasets.

Less importantly, I am not convinced that the community needs another medium-scale dataset of mesh-based physics simulations, and I struggle to see in what DeformingBeam differs from other existing tasks.

**Minor** (little to no effect on my review)
- The author used "dynamic" in several sentences to describe their clustering method. Yet, it is not based on "dynamic" (it is not taking time into account, which is what dynamics is about). I think the words "physical" and "dynamic" have been used interchangeably, but it is confusing for me (some sample lines where dynamics is used arguably misleadingly: line 78/79, line 183, line 216, ...)
- The related work section could benefits from few words about existing datasets for physics on meshes, since DeformingBeam is one of the contributions. In general, while the related work section presents well the state-of-the-art, it could better situate your contribution in the existing literature.
- l.220 "outputs a set OF graph" ?
- l. 806 : "we use the its shortest distance"
- l. 433 : "simulation.(SPACE)More"

**Motivation for my grade**
My rating is based on the lack of significant contributions of the paper. Most of the architecture is based on EAGLE. The only noticeable novelty is the better clustering technique, but the paper only presents experimental results on some datasets and fails to extract general knowledge that would benefit the community.
I strongly encourage the authors to (1) prove me wrong if the contribution extends beyond the clustering technique and (2) strengthen the paper with a better analysis of the results, including when and why the proposed model performs better.

**Questions:**

- You tested several segmentation methods (i.e. handcrafted features) in table 3. It is not clear what are the conclusions of this experiment, since some methods work better on some datasets while some do not, and there is no clear reason why. Could you interpret these results ?
- Did you have any evidence that the segmentation is indeed correlated with the dynamics of the system (and not solely its geometrical description) ?
- Does patterns emerges from the cluster's dynamics that would explain why your method performs well ?
- Did you spotted any behavior of your segmentation method that does not arise from EAGLE's purely geometric clusters that would explain or give an intuition about how the model benefits/uses these clusters ?
- You mention in the introduction that previous work "face drawbacks like manual effort or inaccurate mesh edges". Can you please develop why your method does not suffer from the same issues ?
- While your method is (I think) very close from EAGLE, why not evaluating on the corresponding dataset, which has been designed specifically for very similar models ? It has proven to be sensibly harder than CylinderFlow which exhibits a lot of regularities.
- Where did you sampled the initial condition that you provided to the models ? Is it randomly sampled in a longer simulation or does it corresponds to the very first timestep ? If not, it would be interesting to compare the models in a realistic scenario where the simulation is done from a "cold start".
- I am very confused by the description of the physics-guided segmentation (line 244 to 265). First, you mention that you apply METIS to obtain clusters, followed by SLIC to refine them (to be honest, I'm not familiar with these algorithms, I had to look them up), but it seems that you only used algorithm 1 (which seems to be a K-means with handcrafted features). Am I mistaken ? Is there a difference between your proposed approach and a K-means algorithm ? I think this could be made clearer in the paper.
- Line 225 to 230, you introduce a notation for overlapping clusters. This is not used in the main paper, and the ablation shows that this has negative effect on two datasets and positive on the last one. Why so ? Can we generalize from this experiment to understand which property of the underlying physics could benefits from $\delta>0$ ?

# Rebuttal
The authors provided a thorough and detailed reply to my review, accompanied by a substantial amount of supplementary results and additional analyses. These additions offer greater insight into how the contribution improves upon the state-of-the-art compared to Eagle.

As mentioned in my original review, I believe this is an interesting and valuable line of research. However, the paper contains numerous unsupported claims. For instance, the title refers to "physics-aligned" clustering, but the method does not incorporate any physics priors beyond basic geometric considerations. The rebuttal has addressed some of my concerns, but the main paper remains highly misleading and, in my opinion, poorly structured.

I have raised my score from 3 to 5 but still recommend rejection. The rebuttal lays the foundation for significant improvement, but the main paper should be thoroughly restructured to reflect these changes, particularly regarding the claims and findings.

---

> ### Author Response · Authors · 2024-11-22
> **Response to Reviewer YvcY (part 1)**
>
> We sincerely appreciate the reviewer for taking the time to review our paper and for providing valuable, in-depth questions and feedback. Below, we provide detailed responses to the questions and comments raised.
> > Q1: State contribution of the work extends beyond the clustering technique
>
> We thank the reviewer for thoughtful review and for expressing concerns regarding the contributions of our paper. We appreciate the opportunity to clarify the distinctions between our method and EAGLE and to highlight the unique contributions our work brings to the field. While our approach shares some conceptual similarities with EAGLE, there are significant differences that constitute substantial contributions and lead to improved performance. Please see the summarized contributions of our work below:
> 1. **Methodological Innovations**
>
>    (a) **Novel Physics-Guided Segmentation Method:**
>       - [*Physically Meaningful Segments*]: Introduced a novel clustering algorithm that incorporates physics-informed features into the segmentation process. Unlike EAGLE's fixed-size, node-centric clustering, our method adaptively adjusts segment shapes and locations based on mesh topology and physical priors (e.g., distances to boundaries and obstacles), resulting in segments that align with the underlying dynamics.
>       - [*Superior Performance with Fewer Segments*]: Achieved superior performance without the need for a large number of segments by leveraging physics-guided segmentation. Thoroughly analyzed the effect of segment number on performance, providing insights into how segmentation granularity impacts model accuracy and predicted mesh quality—an aspect not discussed by EAGLE.
>
>    (b) **Segment Feature Aggregation Mechanism:**
>       - [*Simplified Aggregation with Average Pooling*]: Reduced model complexity by using average pooling for segment feature aggregation, which is invariant to segment size, unlike EAGLE's GRU-based method.
>       - [*Avoidance of Sequence Dependency*]: Avoided issues related to sequence dependency and inconsistencies due to random node ordering inherent in GRU-based methods, enhancing robustness and efficiency.
>
>    (c) **Meaningful Macro-Level Information Exchange:**
>       - [*Physics-Informed Attention Mechanism*]: Exchanged higher-level information through a physics-informed attention mechanism, as opposed to EAGLE's attention over numerous small clusters with less physical significance.
>       - [*Efficient Information Transfer*]: Enabled more meaningful and targeted macro-level information exchange that closely reflects actual physical processes, leading to efficient information transfer and improved modeling of long-range interactions.
>
>    (d) **Extended Analysis and Enhancements:**
>       - [*Incorporation of Positional Encoding (PE)*]: Enhanced network expressivity by incorporating positional encoding and thoroughly analyzed its impact under various conditions.
>       - [*Investigation of Overlapping Segments*]: Examined the benefits of overlapping segments, gaining insights into scenarios where overlaps enhance performance.
>
> 2. **Applicability and Extension to New Domains**
>
>    (a) **Application to Solid Mechanics and Lagrangian Systems:**
>       - [*Extension Beyond Fluid Dynamics*]: To the best of our knowledge, we are the first to apply clustering methods to solid mechanics problems, particularly those involving long-range interactions and Lagrangian systems with mesh deformation. This extends the applicability of hierarchical GNNs beyond fluid dynamics, which is the primary focus of EAGLE.
>       - [*Addressing Unique Challenges*]: Developed a clustering/segmentation algorithm that integrates well into a hierarchical framework for deformable Lagrangian meshes, providing a method applicable to a wide range of dynamic systems, whether fluid or solid.
>
> 3. **Robustness and Scalability**
>
>    (a) **Handling Mesh Irregularity and Scalability:**
>       - [*Flexibility in Segment Size*]: Not constrained by equal-sized clusters or fixed nodes per segment, allowing better handling of varying mesh structures and irregular meshes than EAGLE.
>       - [*Adaptive to Mesh Density*]: Mitigated issues in dense meshes where EAGLE's approach results in too many small clusters by adjusting segments based on mesh density and physical priors, enhancing generalization to larger and more complex systems.
>      - [*Generalizabiliy to Larger-scale Dataset*]: Demonstrated ability to effectively generalize to larger-scale datasets, which is essential for real-world simulation applications."
>
> (Response to Q1 to be continued)

---

> ### Author Response · Authors · 2024-11-22
> **Response to Reviewer YvcY (part 2)**
>
> (Continuation of the response to Q1)
>
> 4. **Comprehensive Evaluation and Insights**
>
>    (a) **Comprehensive Evaluation Across Metrics:**
>       - [*Incorporation of Mesh Quality Metrics*]: Considered multiple metrics to evaluate the quality of the predicted meshes. Mesh quality is crucial for Lagrangian systems because the mesh deforms along with the material; maintaining high-quality meshes is essential for accurately capturing dynamics and ensuring numerical stability over time.
>       - [*Balanced Performance*]: Demonstrated that our method achieves an excellent balance across prediction accuracy, computational efficiency, and mesh quality—being the first to consider mesh quality in this context.
>
>    (b) **Comprehensive Comparison of Hierarchical Designs:**
>      - To the best of our knowledge, we are the first work to comprehensively compare hierarchical models for dynamic system prediction, including both multi-level structures (e.g., g-U-Net, BSMS-GNN) and cluster-based structures (e.g., EAGLE and our MMSGN). This comparison evaluates various metrics such as prediction accuracy, predicted mesh quality, and computational efficiency, offering valuable insights for future research.
>
> 5. **Computational Efficiency and Real-World Applicability**
>
>    (a) **Reduced Computational Burden:**
>       - By using fewer, physics-informed segments, we reduce computational burden compared to EAGLE, which requires many small segments, making our method more efficient and applicable to real-world problems.
>
>    (b) **Applicability to Real-World Problems:**
>      - Our method's efficiency and ability to handle complex, irregular meshes make it more applicable to real-world problems where computational resources and scalability are critical.
>
> 6. **Newly Introduced DeformingBeam Dataset**:  (please refer to our response to Q10 for details)
>
>     (a) **Challenging Long-Range Interactions:**
>       - Our dataset includes complex long-range interactions, providing a rigorous test for model performance under scenarios susceptible to over-smoothing.
>
>     (b) **Diverse Mesh Structures and Interaction Scenarios:**
>       - It features a variety of mesh configurations and diverse interaction scenarios between beams and obstacles, enhancing the dataset's complexity and applicability.
>
>     (c) **Standard and Scaled-Up Versions:**
>      - To the best of our knowledge, this is the first dataset to offer both a standard and a scaled-up version, making it a valuable resource for directly evaluating model scalability.
>
>
> > Q2: Strengthen the paper with a better analysis of the results, including when and why the proposed model performs better.
>
> We appreciate the reviewer’s suggestions and have incorporated several in-depth analyses of our method. Below is a high-level summary of the additions made to the paper:
> - We added two additional metrics-Chamfer Distance and Aspect Ratio (details in Appendix D.1)-for a more comprehensive assessment of the predicted meshes, updating Tables 3 to 7 and using them to guide our ablation studies detailed in Appendix E.2.
> - We introduced three additional metrics—Conductance, Edge Cut Ratio, and Silhouette Score (details in Appendix D.2)—to thoroughly evaluate the segmentation methods by assessing both inter-segment and intra-segment qualities.
> - To understand how predicted mesh properties in each segment vary over time for different methods, we included additional visualizations in Figure 6 and analysis in Appendix D.3, where mesh nodes are colored based on the average prediction error within their segments.
> - We analyzed the correlation between these segmentation metrics and overall dynamic system performance, including mesh quality and prediction error, as shown in Figure 9(a–c) and Appendix E.2.
> - To comprehensively evaluate the effect of segment number and determine the optimal number of segments for a given dataset, we analyzed prediction accuracy across a wide range of segment counts during training (mainly on the DeformingBeam dataset), as shown in Figure 9(d) and Appendix E.2.
> - We conducted a comprehensive analysis of the effects of position encoding and segment overlap on model performance under different segment counts across all three datasets, as shown in Figure 10 and Appendix E.2.

---

> > ### Author Response · Authors · 2024-11-22
> > **Response to Reviewer YvcY (part 3)**
> >
> > > Q3: You tested several segmentation methods and it is not clear what are the conclusions of this experiment. Could you interpret these results? Explain why and how the contribution actually benefits the learning of the dynamics.
> > - To thoroughly evaluate the different segmentation methods, we first introduced three additional metrics: Conductance, Edge Cut Ratio, and Silhouette Score. These metrics are crucial for assessing both the inter-segment and intra-segment qualities of the mesh partitions, providing a comprehensive understanding of each method's effectiveness. The details of each metrics are described in Appendix D.2.
> > - We then analyzed how these segmentation metrics correlate with overall dynamic system performance, such as mesh quality and prediction accuracy. This allowed us to interpret how different segmentation methods impact the learning of dynamics  (shown in Figure 9(a-c)). Our findings indicate that segmentation methods incorporating physics-informed features—particularly those using both obstacle and boundary distances with exponential transformations—generally lead to better model performance across different datasets. This suggests that:
> >
> >     - [*Alignment with Dynamics*]: Segmentation that reflects physical influences helps the model learn the system's dynamics more effectively.
> >     - [*Enhanced Segment Quality*]: Improved intra-segment cohesion and minimized inter-segment interactions facilitate better learning of localized patterns.
> >     - [*Benefit to Learning*]: By emphasizing critical regions through exponential transformations, the model focuses on areas with significant dynamic changes, enhancing prediction accuracy.
> > - These results demonstrate that the choice of segmentation method significantly affects the model's ability to learn dynamic behaviors. Introducing additional metrics allowed us to evaluate segmentation quality comprehensively, revealing that physics-informed segmentation benefits the learning of dynamics by aligning mesh partitions with the system's inherent physical properties.
> >
> > > Q4: (merged question) Is there evidence that your segmentation method is indeed correlated with the dynamics of the system (and not solely its geometrical description), and how does this benefit the learning of the dynamics compared to EAGLE's purely geometric clusters? Can you explain how patterns emerging from your clusters' dynamics contribute to your method's improved performance?
> > -  To demonstrate that our segmentation method is correlated with the dynamics of the system rather than just its geometrical structure, we conducted detailed analyses and visualizations {(please refer to Figure 6 and Appendix D.3)}. In Figure 6, we visualized simulation rollouts over time, coloring each node based on average prediction error within its segment. This allowed us to observe how segments correspond to dynamic behaviors. Below are our main observations:
> >
> >     - [*Uniform Dynamic Behavior within Segments*]: In our method, segments exhibit uniform colors across time steps, indicating that nodes within the same segment share similar dynamic behaviors. This uniformity suggests that our segmentation groups regions experiencing similar physical interactions, aligning with the system's dynamics.
> >     - [*Capturing Dynamic Patterns*]: For the Cylinder Flow dataset, which exhibits periodic wake patterns behind the cylinder, our segmentation aligns with these flow patterns. Segments maintain consistent shapes and colors that correspond to the evolving dynamics, effectively capturing the temporal and spatial patterns of fluid flow.
> >     - [*Symmetry Preservation*]: In cases with symmetrical dynamics occur, such as some cases in DeformingBeam, our method produces symmetric segments with similar dynamic properties. This is evident from the consistent coloring of symmetric segments, reflecting that our segmentation respects the physical symmetry and associated dynamics.
> > - Comparing with EAGLE, Figure 6 shows that EAGLE's segments often have adjacent segments with significantly different colors, indicating that the clustering fails to capture consistent dynamic behaviors within physically correlated regions, leading to less cohesive segmentation and poorer alignment with the system's underlying dynamics. Also, the figures show that EAGLE's segmentation does not capture dynamic patterns such as periodic wakes or symmetrical behaviors, suggesting a lack of correlation with the system's dynamics. Moreover, according to our additional quantitative analysis in Figure 9(a-c), our segmentation method yields better performance across various metrics compared to EAGLE. These improvements indicate that our segments are more cohesive and aligned with dynamic behaviors, enhancing the model's ability to learn and predict system dynamics effectively.

---

> > > ### Author Response · Authors · 2024-11-22
> > > **Response to Reviewer YvcY (part 4)**
> > >
> > > > Q5: You mention in the introduction that previous work "face drawbacks like manual effort or inaccurate mesh edges". Can you please develop why your method does not suffer from the same issues?
> > > - Unlike existing methods that either down-select nodes and reconstruct mesh edges (Liu et al., 2021; Li et al., 2020c; Cao et al., 2023)—disrupting the original mesh structure and losing critical spatial relationships—or disregard edge information entirely during clustering, as in EAGLE (Janny et al., 2023), our approach preserves the original connectivity by leveraging METIS for initial segmentation. METIS ensures consistent and reproducible clusters given the same mesh and configuration, maintaining the integrity of mesh edges and spatial relationships. Additionally, our Physics-Guided Segmentation refines these segments using physics-informed features without altering the original mesh structure. This guarantees accurate dynamics propagation and retains spatial significance, avoiding the inaccuracies caused by edge reconstruction in other methods.
> > > - Furthermore, many existing approaches rely on manual effort, such as manually drawing coarser meshes for the original geometry (Fortunato et al., 2022). In contrast, our method eliminates manual intervention by automating the segmentation process with physics-guided features, computed directly from mesh geometry and system properties. This fully automated framework, including METIS-based initialization and adaptive refinement, ensures scalability and applicability to diverse dynamic systems. As a result, our approach not only overcomes the challenges of inaccurate mesh edges but also avoids the labor-intensive efforts required in prior methods, leading to more reliable, efficient, and generalizable simulations.
> > >
> > > > Q6: Why not evaluating on the corresponding dataset in EAGLE, which has been designed specifically for very similar models ? It has proven to be sensibly harder than CylinderFlow which exhibits a lot of regularities.
> > >
> > > We thank the reviewer for thoughtful question and we acknowledge that it would be more beneficial to demonstrate the effectiveness of our method by using EAGLE's dataset for direct comparison. However we did not evaluate our method on the EAGLE dataset initially for several reasons:
> > > - [*Dataset Characteristics*]: Although CylinderFlow has certain regularities, the EAGLE dataset has its own limitations, such as large regions with small velocity, making the learning task simpler in many areas. Also, despite CylinderFlow's regularities, EAGLE only achieved a modest 7.2\% improvement over MGN on this dataset, whereas our method demonstrated a significant 39\% improvement. This indicates that our approach effectively captures dynamics even in CylinderFlow datasets and would likely outperform EAGLE on its own dataset.
> > > - [*Focus on Diverse Systems and Mesh Types*]: While EAGLE focuses exclusively on fluid dynamics with triangular meshes, we aim to explore broader applications, including solid mechanics systems (e.g., DeformingPlate and DeformingBeam) with Lagrangian formulations and diverse mesh types (tetrahedral, prism). This provides a more comprehensive evaluation of our method's versatility across physical systems.
> > > - [*Future Evaluation*]: Given our method’s strong performance, we believe it would perform well on the EAGLE dataset. However, our current focus was to validate its generality across a wide range of dynamics and mesh configurations and we'd like to include evaluations on EAGLE in future work for direct comparison.
> > >
> > > > Q7: Where did you sampled the initial condition that you provided to the models ? Is it randomly sampled in a longer simulation or does it corresponds to the very first timestep ? If not, it would be interesting to compare the models in a realistic scenario where the simulation is done from a "cold start".
> > >
> > > We indeed consider starting from the very first time step during testing and perform rollouts thereafter. Specifically the evaluation is done in two ways:
> > > - [*Realistic Scenario Rollouts*]: Both the RMSE-50 and RMSE-all metrics correspond to average RMSE values for 50-step rollouts and entire trajectory rollouts, respectively. These rollouts begin from the first time step and ensemble a realistic scenario where the simulation starts from a "cold start" and progresses sequentially.
> > > - [*Sliding Window Rollouts*]: The RMSE-1 metric corresponds to the RMSE for a 1-step rollout. In this evaluation, the initial condition begins at the first time step, and the model predicts the subsequent step. The sliding window then advances by one time step, using the second time step to predict the third step, and continues this process throughout the trajectory. This approach assesses the model’s ability to make localized predictions at any point along the trajectory.

---

> > > > ### Author Response · Authors · 2024-11-22
> > > > **Response to Reviewer YvcY (part 5)**
> > > >
> > > > > Q8: Clarification on the physics-guided segmentation method
> > > > - The original SLIC algorithm is a popular method for generating visually homogeneous segments in an image. It works by clustering image pixels in a 5D space defined by their color and spatial coordinates, using a customized k-means clustering approach. Here, we extend SLIC to mesh graph by using physics-guided features and spatial coordinates in clustering. To initialize the clusters, we apply METIS to create balanced graph partitions. We have updated Algorithm 1 to include pseudo-code for the METIS partitioning used in the initialization stage, enhancing the clarity and understanding of the overall algorithm.
> > > >
> > > > > Q9: Add detailed analysis on overlapped segment $\delta$. Can we generalize from this experiment to understand which property of the underlying physics could benefits from $\delta > 0$?
> > > >
> > > > - We conducted a comprehensive analysis of the impact of $\delta$, detailed in Figure 10(b) and Appendix E.2. To provide a more thorough evaluation and better guide our analysis, we introduced two additional metrics, described in Appendix D.1. According to the results, the effectiveness of adding overlap between segments ($\delta$ > 0) depends on both the segment count and the characteristics of the dataset, such as dimensionality, mesh type, and system dynamics. Overlapping segments are more beneficial with higher segment counts where discontinuities are more prevalent. In Eulerian systems, overlaps enhance the capture of complex interactions and smooth transitions on fixed meshes, leading to improved representation of fluid dynamics. Conversely, in Lagrangian systems where meshes move with the material, overlaps can create redundancy and complicate connectivity, with their impact on model performance varying based on mesh structures and deformation behaviors. For example, in the Deforming Beam dataset, which uses a prism mesh suited for directional deformation, overlapping segments improve performance by facilitating smooth transitions along its mesh surface, especially with a higher number of segments. In contrast, the Deforming Plate dataset employs a tetrahedral mesh with complex, isotropic deformations, where overlaps introduce unnecessary complexity and redundancy, resulting in decreased performance. Therefore, despite both being 3D Lagrangian systems, the different mesh types and deformation patterns explain why overlapping segments benefit the Deforming Beam but not the Deforming Plate.
> > > >
> > > > > Q10: In what aspect does DeformingBeam differs from other existing dataset?
> > > >
> > > > We created the DeformingBeam dataset for three primary reasons:
> > > > - [*Challenging Long-Range Interactions*]: DeformingBeam exhibits significant long-range interactions, characterized by the largest graph diameter among the three datasets we utilized (see Table 2). This makes it particularly challenging and ideal for evaluating model performance under potential over-smoothing conditions, a common issue in mesh-based physics simulations. Existing datasets often lack this level of complexity, limiting their effectiveness in testing advanced models.
> > > > - [*Diverse Mesh Structures and Interaction Scenarios*]: The DeformingBeam dataset comprises a variety of beam mesh structures and diverse interaction scenarios between beams and obstacles. This diversity introduces complexities that existing datasets may not adequately capture. For example, as shown in Table 1, several state-of-the-art methods (e.g., MGN, BSMS, EAGLE) demonstrate similar RMSE-all errors on DeformingBeam, indicating their limitations in handling the dataset's complexity. In contrast, our method achieves over a 50\% performance improvement to the second-best model, highlighting its superior ability to manage diverse and challenging scenarios that mirror real-world applications.
> > > > - [*Scalability with Scaled-Up Version*]: We introduce DeformingBeam(Large), a scaled-up version with twice the graph diameter of the standard DeformingBeam dataset. To the best of our knowledge, this is the first dataset to provide both a standard and a scaled-up version, serving as a valuable resource for directly evaluating model scalability. Scalability is crucial for real-world applications, where systems are often extremely large, and collecting sufficient training data at such scales is impractical. DeformingBeam and its scaled-up counterpart, DeformingBeam(Large), allow researchers to test whether their models can effectively scale from small-scale to large-scale systems, closely simulating real-world implementation scenarios.

---

> > > > > ### Author Response · Authors · 2024-11-22
> > > > > **Response to Reviewer YvcY (part 6)**
> > > > >
> > > > > > Q11: I think the words "physical" and "dynamic" have been used interchangeably, but it is confusing for me.
> > > > > - We thank the reviewer for the detailed comments and agree that it may cause confusion. We have made the necessary adjustments in the relevant sections of the paper to ensure clear and accurate terminology throughout.
> > > > >
> > > > > > Q12: The related work section could benefits from few words about existing datasets for physics on meshes.
> > > > > - We thank the reviewer for the constructive suggestion and we have included a discussion on existing datasets for unstructured mesh-based simulations in Appendix A.1.
> > > > >
> > > > > > Q13: Typos: Line.220: "outputs a set OF graph" ? Line. 806 : "we use the its shortest distance"; Line. 433 : "simulation.(SPACE)More".
> > > > > - We appreciate the review for the careful reading of the manuscript. We have corrected the typos in the revised version of our paper.

---

> ### Comment · Reviewer_YvcY · 2024-11-26
> **Reply**
>
> Thank you for your comprehensive reply. I appreciate the time you took to address my review but kindly suggest keeping replies as concise as possible. That said, I understand that long, negative reviews often call for detailed responses. Since some of your replies overlap, I will only address specific points.
>
> **Explicit list of contributions**
> I appreciate that you clarified your contributions, particularly in comparison to Eagle. However, I still have concerns about the novelty, and some of the claims in your rebuttal are not well supported.
>
> 1. a) Your clustering method is indeed more relevant than the naive approach in Eagle and has a clear impact on performance. However, it is not physics-guided. The introduced features, while logical, are purely geometric (e.g., distance to obstacles/boundaries) and do not incorporate any explicit physical knowledge.
>     b) Some of your claims are very misleading. Your attention mechanism is not physics-informed; it is neither constrained by nor aware of the underlying physics. For example, you assert that Eagle clusters have "less physical significance," but this is unsupported, and the fact that your model outputs better prediction is not a proof of the physical significance of the cluster. Additionally, you state that the information exchange is "more meaningful." What does this mean, and which experiment demonstrates the "meaningfulness" of this exchange?
> 2. Why would equal-sized clusters improve the handling of irregular meshes? You claim that your method "enhances generalization to larger and more complex systems," but where is the evidence to support this?
>
> I acknowledge the other points of novelty mentioned in your response and agree with them, but these do not seem to constitute the core claims of the paper.
>
> **Added experiments on mesh quality and correlation with performance**
> Thank you for including this impressive set of new results. I found these experiments extremely interesting in many aspects. Visual inspection of the clusters in Fig. 6 shows that they are less biased toward circular shapes, which makes a lot of sense for CylinderFlow. Furthermore, the results in Fig. 9 demonstrate a clear correlation between performance and geometric mesh quality metrics.
>
> Overall, these results significantly strengthen the paper and, in my view, should have been central to its original message.
>
> **Comparison with Eagle dataset**
> I understand your reasoning, but given that your paper is strongly inspired by Eagle while aiming to address its weaknesses, a comparison with the Eagle dataset seems mandatory. Relying on performance on other datasets to argue that your method would outperform Eagle on its dataset as well is not well supported.
>
> I edited my review and updated my score and its motivation.

---

> > ### Author Response · Authors · 2024-11-28
> > **Follow-up Response to Reviewer YvcY (part 1)**
> >
> > We appreciate the reviewer's valuable feedback and are glad that the reviewer acknowledged that our additional results have significantly strengthened the paper. We also acknowledge the reviewer's suggestion to keep our replies concise and will strive to be more succinct in future communications. Please find our responses to the reviewer's follow-up questions below; we hope these will clarify any remaining concerns:
> > > Q1: Your clustering method is not physics-guided. The introduced features, while logical, are purely geometric (e.g., distance to obstacles/boundaries) and do not incorporate any explicit physical knowledge.
> >
> > We thank the reviewer for their insightful feedback and we appreciate the opportunity to further clarify the points the reviewer raised. We would like to answer this in two aspects:
> > - While it's true that the features we introduced are geometric related, these geometric characteristics are intrinsically linked to physical properties in many physical systems. For example:
> >     - [*Boundary Layer Effects*] : The distance to a boundary influences the development of boundary layers, which in turn affects velocity profiles, shear stresses, and overall flow behavior.
> >     - [*Wake Formation*]: The geometry of an object influences the wake region formed behind it, affecting downstream flow properties and turbulence levels.
> >     - [*Stress Concentration*]: The presence of holes, notches, or abrupt changes in cross-sectional area can lead to localized stress concentrations, directly affecting how the material will deform or fail under load.
> >     - [*Contact Mechanics*]:  The shape of contacting surfaces affects stress distributions, wear rates, and potential for material deformation at the contact interface.
> >
> >   These examples illustrate that geometric features often govern physical phenomena. Therefore, incorporating geometry-related information can indeed make the clustering method physics-guided.
> > - Our proposed clustering algorithm is designed to be flexible and can readily incorporate various physical properties, reinforcing its classification as physics-guided mesh segmentation. For instance:
> >     - [*Permeability Maps*]: Different porosity levels affect fluid storage capacity and flow paths, making tailored mesh segmentation beneficial.
> >     - [*Viscosity Variations*]: In fluids with temperature-dependent viscosity or in multiphase flows, known viscosity changes can guide mesh refinement in critical regions.
> >     - [*Material Property Distributions*]: In multi-material components, regions with varying elastic moduli, Poisson's ratios, or yield strengths require careful mesh consideration.
> >     - [*Residual Stress Fields*]: Pre-existing stress distributions from manufacturing processes can affect how loads are transferred through a structure.
> >
> >   Then, we can incorporate functions based on these field variables into our segmentation algorithm. Exemplar functions are:
> >     - [*Property-Based Weighting Functions*]: Assigning weights to mesh elements based on physical properties (e.g., higher weights in regions with significant material property changes) to influence clustering.
> >     - [*Adaptive Thresholding*]: Defining thresholds for physical properties to identify regions that require finer mesh resolution.
> >     - [*Gradient Analysis Functions*]: Using gradients of known property fields (e.g., changes in porosity or material stiffness) to detect interfaces or zones with significant property variations.
> >
> >   However, as the existing datasets lack comprehensive prior information, it is challenging to test more advanced physics priors. Nonetheless, by demonstrating the effectiveness of geometric-based physics priors, we have shown that our framework can leverage fundamental physical insights. We anticipate that incorporating more complex physics priors will further improve the performance. Furthermore, we plan to explore modal analysis techniques in engineering and fracture mode analysis in graphics as potential avenues to advance physics-driven mesh segmentation methods in our future work.

---

> ### Author Response · Authors · 2024-11-28
> **Follow-up Response to Reviewer YvcY (part 2)**
>
> > Q2: Your attention mechanism is not physics-informed and the fact that your model outputs better prediction is not a proof of the physical significance of the cluster.
> - We apologize for any confusion our previous statement may have caused and appreciate the opportunity to clarify our points. The reviewer is correct that the attention mechanism itself is not inherently physics-informed and has no direct awareness of the underlying physics. What we intended to convey is that, in our method, the attention operates over mesh segments created through physics-guided segmentation. Consequently, the attention mechanism indirectly leverages the underlying physics by aggregating information across these physically significant regions.
> - To demonstrate that a mesh clustering or segmentation method has physical significance, it is important to show that the clusters correspond to meaningful physical properties, behaviors, or phenomena within the system we are studying. We believe that our additional results (e.g., Figures 6 and 9) support this point by illustrating that our segmentation method not only enhances predictive accuracy but also produces clusters with greater physical significance. This is achieved by effectively capturing temporal and spatial dynamics that are aligned with the system's physics, thereby outperforming EAGLE in this regard.
> - We sincerely value the reviewer's feedback and are open to any suggestions the reviewer may have regarding additional experiments or analyses that could further support this statement. We are more than willing to incorporate the reviewer's recommendations to strengthen our work.
>
> > Q3: You state that the information exchange is "more meaningful." What does this mean, and which experiment demonstrates the "meaningfulness" of this exchange?
>
> When we say that the information exchange in our model is "more meaningful," we mean that it more effectively and accurately captures the essential physical interactions and dynamics of the system being modeled. This improvement arises because our model's information exchange is based on mesh segments that align with key physical regions and behaviors within the system. As a result, the model achieves better predictive accuracy and generalization by effectively capturing and leveraging critical physical interactions. Below are experiments that demonstrate the "meaningfulness" of the exchange:
> - [*Visualization of Segmentation Alignment*]: (Figure 6) The segments produced by our physics-guided segmentation align with important physical regions in the system. This alignment allows the model to capture essential interactions more effectively, as it focuses on areas where significant physical processes occur.
> - [*Correlation Between Segmentation Quality and Model Performance*]: (Figure 9) We observed that higher-quality segments, which better represent the physical structure of the system, lead to improved predictions. This correlation indicates that when the segmentation accurately reflects the system's physical properties, the information exchange becomes more effective, enhancing the model's performance.
> - [*Performance Improvements Over State-of-the-art Methods*]: (Tables 1 and 3; Figure 3, 4, and 13-16) Our model outperforms existing methods in terms of prediction accuracy and other evaluation metrics. This superior performance suggests that the information exchange facilitated by our physics-guided segmentation leads to more accurate modeling of the system's dynamics, reinforcing the effectiveness of our approach.
>
> We appreciate the reviewer's insightful question and are happy to provide further clarification. If the reviewer has any suggestions for additional experiments or analyses that could better demonstrate the meaningfulness of our information exchange, we are more than willing to consider them to strengthen our work.
>
> > Q4: Why would equal-sized clusters improve the handling of irregular meshes?
>
> - We thank the reviewer for the question. However, there appears to be a misunderstanding and we are happy to clarify it. We did NOT claim that equal-sized clusters improve the handling of irregular meshes. In contrast, equal-sized clusters, like those used in EAGLE, are not effective for irregular meshes. They impose uniformity that does not adapt to varying node densities and complex geometries present in irregular meshes. In contrast, our method allows clusters to vary in size and shape, adapting to the mesh's inherent variability and aligns with the system's physical dynamics. This adaptability enhances the model's ability to handle irregular meshes and improves predictive accuracy.

---

> > ### Author Response · Authors · 2024-11-28
> > **Follow-up Response to Reviewer YvcY (part 3)**
> >
> > > Q5: You claim that your method "enhances generalization to larger and more complex systems," but where is the evidence to support this?
> >
> > - We've provided evidence of enhanced generalization to larger and more complex systems through experiments on our DeformingBeam(Large) dataset. This dataset is larger and more complex because it has twice the graph diameter and significantly more nodes than the original DeformingBeam dataset, introducing longer-range interactions and increased complexity in mesh structures. As shown in Figure 4(c) and Table 7, our method consistently achieves the lowest prediction error and mesh continuity error when trained on the smaller-scale DeformingBeam dataset and tested on DeformingBeam(Large). This demonstrates that our model effectively generalizes from smaller to larger systems, supporting our claim of enhanced generalization to larger and more complex systems.
> >
> > > Q6: Comparison with Eagle dataset
> > - We thank the reviewer for their feedback and agree that directly evaluating our method on the EAGLE dataset is important. We will include experiments on the EAGLE dataset in our revised manuscript to provide stronger evidence supporting our claims. We hope the reviewer understand that we may not be able to complete these experiments before the rebuttal period ends, but we will make every effort to include them as soon as possible. We appreciate the reviewer's suggestion to improve our work.

---

### Official Review · Reviewer_BRQ5 · 2024-11-01

**Soundness:** 3
**Presentation:** 3
**Contribution:** 3
**Rating:** 8
**Confidence:** 3

**Summary:**

This paper presents the Mesh-based Multi-Segment Graph Network (MMSGN), which is a novel hierarchical framework that aims at simulating dynamic systems with high accuracy and computational efficiency. The proposed model uses a two-level interaction mechanism (micro-level local interactions, macro-level global exchanges). By aligning the mesh structure and the physical properties, MMSGN effectively captures both local and long-range dynamics, overcoming common issues in existing mesh-based Graph Neural Networks (GNNs), such as oversmoothing and excessive computational load. The paper validates MMSGN on multiple datasets, showing that it outperforms several state-of-the-art methods in terms of accuracy, mesh quality, and efficiency, and introduces a new dataset (DeformingBeam) for evaluating mesh-based simulations.

**Strengths:**

- The combination of micro- and macro-level exchanges is interesting, innovative, and well-motivated. It also aligns with properties of real-world dynamic systems.
- The experiemnts indicate that MMSGN is able to achieve a balance of high prediction accuracy and computational efficiency.
- The approach has been appropriately validated w.r.t. to different experiments and datasets.
- The paper is well-written and easy to follow. Sections are meaningfully structured and enough details are provided.
- The paper introduces a new dataset (DeformingBeam), which provides an additional benchmark for future research.

**Weaknesses:**

- While the hierarchical approach appears conceptually sound, MMSGN may be challenging to re-implement due to its multi-level structure.
- The paper briefly mentions comparisons with other state-of-the-art methods but would  benefit from a more in-depth analysis (e.g. additional benchmarks) to further explore the capabilities and limitation sof MMSGN.
- Although the paper highlights scalability, it is unclear whether this extends to systems with highly irregular or extremely large-scale meshes. More details would be appreciated on this.
- Not enough details are provided on how to incorporate diverse boundary conditions. A more in-depth discussion would be beneficial.

**Questions:**

- How does MMSGN handle complex boundary conditions?
- Could it effectively simulate systems where boundaries have non-standard behaviors or physical constraints?
- Has the model been tested on highly irregular or unstructured meshes, and if so, how does it perform in these cases?
- Can the authors provide a more detailed analysis of the computational complexity of MMSGN?
- Given its design, could MMSGN be applied to domains beyond traditional physics-based simulations (e.g. social network dynamics)?
- The authors may also want to add the following paper to the list of related work: H. Shao, T. Kugelstadt, T. Hädrich, W. Pałubicki, J. Bender, S. Pirk, D. L. Michels, Accurately Solving Rod Dynamics with Graph Learning, Conference on Neural Information Processing Systems (NeurIPS), 2021

**Details Of Ethics Concerns:**

No ethical concerns.

---

> ### Author Response · Authors · 2024-11-22
> **Reply to Reviewer BRQ5 (part 1)**
>
> We sincerely thank the reviewer for taking the time to review our paper and provide valuable feedback. Below, we address the questions and comments raised in detail:
> > Q1: While the hierarchical approach appears conceptually sound, MMSGN may be challenging to re-implement due to its multi-level structure.
> - We acknowledge the concern regarding the potential complexity of re-implementing MMSGN due to its multi-level structure. In fact, the hierarchical design of MMSGN is modular and adaptable, making it straightforward to apply to various dynamic systems. To further alleviate any implementation challenges, as mentioned in the paper, we will make the complete code publicly available upon the paper’s acceptance. This accessibility will enable researchers to easily utilize and extend our framework for their specific applications.
>
> > Q2: The paper briefly mentions comparisons with other state-of-the-art methods but would benefit from a more in-depth analysis to further explore the capabilities and limitations of MMSGN.
>
> We appreciate the reviewer’s suggestions and have incorporated several in-depth analyses of our method. Below is a high-level summary of the additions made to the paper:
> - We added two additional metrics-Chamfer Distance and Aspect Ratio (details in Appendix D.1)-for a more comprehensive assessment of the predicted meshes, updating Tables 3 to 7 and using them to guide our ablation studies detailed in Appendix E.2.
> - We introduced three additional metrics—Conductance, Edge Cut Ratio, and Silhouette Score (details in Appendix D.2)—to thoroughly evaluate the segmentation methods by assessing both inter-segment and intra-segment qualities.
> - To understand how predicted mesh properties in each segment vary over time for different methods, we included additional visualizations in Figure 6 and analysis in Appendix D.3, where mesh nodes are colored based on the average prediction error within their segments.
> - We analyzed the correlation between these segmentation metrics and overall dynamic system performance, including mesh quality and prediction error, as shown in Figure 9(a–c) and Appendix E.2.
> - To comprehensively evaluate the effect of segment number and determine the optimal number of segments for a given dataset, we analyzed prediction accuracy across a wide range of segment counts during training (mainly on the DeformingBeam dataset), as shown in Figure 9(d) and Appendix E.2.
> - We conducted a comprehensive analysis of the effects of position encoding and segment overlap on model performance under different segment counts across all three datasets, as shown in Figure 10 and Appendix E.2.
>
> > Q3: How does MMSGN handle complex boundary conditions?
> - MMSGN is fully data-driven and learns the dynamics directly from the simulations, including the imposed boundary conditions. Boundary conditions are incorporated by encoding the node type in the input features, allowing it to distinguish boundary mesh nodes from others. Among the datasets we investigated, CylinderFlow adopts fixed values for boundary nodes; DeformingPlate and DeformingBeam main fixed positions for boundary handle nodes. During rollout, MMSGN explicitly enforces these boundary conditions by assigning the fixed values to the relevant boundary nodes, ensuring consistency with the physical setup.
>
> > Q4: Could it effectively simulate systems where boundaries have non-standard behaviors or physical constraints?
> - MMSGN is capable of learning system dynamics even when non-standard boundary behaviors are involved, as it directly learns the evolution of state variables from ground truth simulations that inherently capture boundary condition effects. However, the current model does not explicitly enforce complex boundary conditions. To address this, future work could integrate strategies from existing studies that focus on incorporating boundary conditions and physical constraints in model design. Notable examples include:\
> [1] *Saad, Nadim, et al. "Guiding continuous operator learning through physics-based boundary constraints." arXiv preprint arXiv:2212.07477 (2022).*\
> [2] *Yang, Shuqi, Xingzhe He, and Bo Zhu. "Learning physical constraints with neural projections." Advances in Neural Information Processing Systems 33 (2020): 5178-5189.*

---

> > ### Author Response · Authors · 2024-11-22
> > **Reply to Reviewer BRQ5 (part 2)**
> >
> > > Q5: Has the model been tested on highly irregular or unstructured meshes, and if so, how does it perform in these cases?
> > - To assess the model's performance on highly irregular or unstructured meshes, we employed variation in element size as a standard metric. Specifically, we categorized the meshes into two groups based on the ratio of the standard deviation of element sizes (e.g., area for 2D meshes, volume for 3D meshes) to the average element size.The lower 50\% represents less irregular meshes, while the upper 50\% comprises more irregular meshes.
> > - As summarized in the table below, our model (MMSGN) consistently outperforms other state-of-the-art models, across all datasets and levels of irregularity. Specifically, in the Cylinder dataset, MMSGN achieves a 20\% reduction in RMSE compared to EAGLE for highly irregular meshes. In the Plate and Beam datasets, MMSGN demonstrates even greater improvements, with RMSE reductions of up to 48.3\% and 56.1\% to the second best model, respectively. Additionally, MMSGN shows significant decreases in Aspect Ratio Error across all irregularity levels, indicating more accurate segment shapes.
> > - These results highlight MMSGN's robustness to mesh irregularity, maintaining superior performance in highly unstructured scenarios where other models experience notable performance degradation. This enhanced performance underscores the effectiveness of our segmentation approach in capturing complex dynamic behaviors, even in challenging mesh configurations.
> >
> > | Dataset   | Model  | RMSE-all (lower 50% irregularity) | RMSE-all (upper 50% irregularity) | Aspect Ratio Error (lower 50% irregularity) | Aspect Ratio Error (upper 50% irregularity) |
> > |-----------|--------|-----------------------------------|-----------------------------------|---------------------------------------------|---------------------------------------------|
> > | **Cylinder** |        |                                   |                                   |                                             |                                             |
> > |           | MGN    | 5.08e-02                          | 3.20e-02                          | -                                           | -                                           |
> > |           | BSMS   | 1.51e-01                          | 9.95e-02                          | -                                           | -                                           |
> > |           | EAGLE  | 7.12e-02                          | 3.14e-02                          | -                                           | -                                           |
> > |           | MMSGN  | **4.06e-02 (↓ 20%)**                  | **2.51e-02 (↓ 20.1%)**                | -                                           | -                                           |
> > | **Plate**    |        |                                   |                                   |                                             |                                             |
> > |           | MGN    | 1.92e-02                          | 3.77e-03                          | 5.24e-03                                   | 5.16e-03                                   |
> > |           | BSMS   | 1.35e-02                          | 7.83e-03                          | 1.27e-02                                   | 2.14e-02                                   |
> > |           | EAGLE  | 4.18e-03                          | 3.48e-03                          | 2.60e-03                                   | 3.37e-03                                   |
> > |           | MMSGN  | **2.16e-03 (↓ 48.3%)**                | **2.78e-03 (↓ 20.1%)**                | **2.53e-03 (↓ 2.69%)**                         | **2.91e-03 (↓ 13.6%)**                         |
> > | **Beam**     |        |                                   |                                   |                                             |                                             |
> > |           | MGN    | 3.69e-04                          | 5.09e-04                          | 8.41e-03                                   | 8.05e-03                                   |
> > |           | BSMS   | 3.86e-04                          | 4.12e-04                          | 1.70e-02                                   | 1.58e-02                                   |
> > |           | EAGLE  | 4.69e-04                          | 3.53e-04                          | 5.13e-03                                   | 4.16e-03                                   |
> > |           | MMSGN  | **2.10e-04 (↓ 43.1%)**                | **1.55e-04 (↓ 56.1%)**                | **3.27e-03 (↓ 36.3%)**                         | **3.11e-03 (↓ 25.2%)**                         |

---

> > > ### Author Response · Authors · 2024-11-22
> > > **Reply to Reviewer BRQ5 (part 3)**
> > >
> > > > Q6: Can the authors provide a more detailed analysis of the computational complexity of MMSGN?
> > > - MMSGN consists of three major components: mesh-based GNN, physics-guided mesh segmentation and mesh segment transformer. Since the physics-guided mesh segmentation is performed only once at the initial time step and can be completed offline, we omit its computational cost for now and focus solely on the computation involved in the mesh-based GNN and the mesh segment transformer. The computational complexity of the GNN is $O(L_1|\mathcal{V}|d^2 + L_1|\mathcal{E}|d^2)$, where $L_1$ is the number of message passing layer, $d$ is the feature dimension, $|\mathcal{V}|$ is the number of mesh nodes and $|\mathcal{E}|$ is the number of mesh edges. The complexity of mesh segment transformer is $O(L_2 K^2 d + L_2 K d^2)$, where $L_2$ is the number of multi-head attention layers, $K$ is the number of segments, and $d$ is the feature dimension. The overall complexity is $O(L_1|\mathcal{V}|d^2 + L_1|\mathcal{E}|d^2 + L_2 K^2 d + L_2 K d^2)$.
> > >
> > > > Q7: Given its design, could MMSGN be applied to domains beyond traditional physics-based simulations (e.g. social network dynamics)?
> > > - Yes. While MMSGN is specifically designed for physics-based simulations, its hierarchical framework—integrating micro-level local interactions with macro-level global exchanges—is adaptable to other fields. In social networks, micro-level interactions can represent direct relationships between individuals or tightly knit groups, while macro-level exchanges capture broader community influences and information spread across distant parts of the network. By replacing mesh segmentation with community detection or clustering techniques tailored to social connections, MMSGN can effectively model both localized and system-wide dynamics in social networks. Additionally, adapting the graph neural network and segmentation modules to incorporate domain-specific features would enhance its ability to capture the unique characteristics of social interactions. Thus, with appropriate modifications, MMSGN’s principles can be successfully applied to social network dynamics and other complex, interconnected domains.
> > >
> > > > Q8: The authors may also want to add the following paper to the list of related work: H. Shao, T. Kugelstadt, T. Hädrich, W. Pałubicki, J. Bender, S. Pirk, D. L. Michels, Accurately Solving Rod Dynamics with Graph Learning, Conference on Neural Information Processing Systems (NeurIPS), 2021
> > > - We thank the reviewer for bringing this valuable work to our attention. We have included it in related work Section 2.1.

---

> > ### Comment · Reviewer_BRQ5 · 2024-11-25
> > **Response**
> >
> > Authors, thank you for responding to my questions in such detail (and also the ones below) and for your patience in waiting to hear back from me. You answers helped me to further understand your method and to clarify my concerns. I am now more convinced about this work and I am willing to increase my score.

---

> > > ### Author Response · Authors · 2024-11-25
> > >
> > > We are pleased that our clarifications effectively addressed the reviewer's concerns and sincerely appreciate the reviewer's decision to raise the score. The constructive feedback provided has been instrumental in refining the paper. Once again, we thank the reviewer for their time and for recognizing the significance of this work.

---

### Official Review · Reviewer_5KUk · 2024-11-04

**Soundness:** 2
**Presentation:** 3
**Contribution:** 2
**Rating:** 5
**Confidence:** 3

**Summary:**

The paper presents the Mesh-based Multi-Segment Graph Network (MMSGN), a model designed to simulate dynamic systems by combining local and long-range information exchanges within a physically aligned hierarchical mesh structure. By segmenting meshes based on physics-informed features, MMSGN efficiently captures complex dynamics and outperforms baseline methods in both prediction accuracy and mesh quality across several datasets. Through empirical analysis, the model demonstrates robust generalization capabilities and scalability, making it suitable for large-scale, complex simulations​.

**Strengths:**

1. **Clarity of High-Level Concept**: The core idea—that deformations primarily remain local to the area of contact and propagate slowly across the entire structure so some clustered feature would be helpful—is intuitive and easy to follow. This perspective makes sense in scenarios where local interactions don’t significantly affect distant areas. I found the paper easy to follow. In addition, I think the paper provides good details on the setup of the experiments (including baselines).

2. **Physics-guided Segmentation**: I like the idea of using segmentation/clustering to "reduce feature space" locally, it draws some interesting connections to modern CNN and ViT structure in CV, and reduced-order modelling in engineering. In addition, Positional Encoding (PE) and Segment Encoding (SE) definitions are grounded in physical intuition by constructions.

3. **Performance**: Surprisingly fast compared to well-established baselines like MGN. However, I would like to learn some intuition behind this -- based on the neural network structure, I can't really tell where the performance boost comes from. Is it because the number of message-passing steps between nodes was reduced?

**Weaknesses:**

1. **Questionable Motivation in Localization**: The emphasis on local deformation may not always hold, especially in fields like computational physics or mechanical engineering, where elasticity often results in global, fast propagating deformation—particularly with low-stiffness materials. This raises concerns that the paper's foundation may not fully align with real-world mechanics.

2. **Limitations in Segmentation Approach**: The paper’s reliance on *purely geometric* (and arguably *topological*) clustering for segmentation may not capture clusters that reflect true mechanical behavior. For instance, with a bird flapping its wings, both wings would oscillate at similar frequencies, we could naturally group them as a single cluster in modal analysis. However, the geometric-only approach used here might yield clusters highly dependent on the initial setup, potentially missing important physical interdependencies. A physics-inspired method, like modal analysis, could provide clusters with greater physical relevance and reduce sensitivity to initialization. This is my major concern with this submission, the current segmentation approach is too *geometric* and already exhibits a **strong** prior on how every deformation must be highly local.

3. **Mesh Resolution Sensitivity and Segment Size Issues**: As number of segments grows, the number of finite elements within them decreases, leading to potential accuracy and performance downgrades. Appendix Table 4 reflects this, with errors increasing as segment size grows in some examples, but the paper lacks clarity on how it determines an optimal cluster count.

**Questions:**

1. **Mesh Continuity**: While the authors use *Mesh Continuity* as the mesh quality benchmark, this metric may not be standard in all visual computing applications or mechanical engineering without proper citations. I would recommend using aspect ratio, or "as-regular-as-possible" metric to measure the uniformness for every element (triangle/tetrahedra) on the mesh, as it is a more common mesh quality metric in FEM literature. Furthermore, the sole use of Hausdorff distance for geometric fidelity could be limiting; adding Chamfer distance would provide a more balanced measure of mesh accuracy.
2. **Underwhelming PE Impact**: Despite the geometric reasoning behind PE(e.g. understanding relative location between within segments), the benefit is marginal (around a 10% reduction in RMSE on the already low error). Could the authors elaborate on the observed impact of PE?
3. **Selection of Optimal Number of Clusters**: The paper includes an empirical analysis in Appendix Table 4, showing that the number of clusters impacts accuracy (with only two data points along the # of clusters dimension shown in the table). Given the algorithm's similarity to K-means, selecting an optimal number of clusters and their initialization will likely be crucial for accuracy and convergence speed. Could the authors expand on their criteria for selecting the optimal number of clusters, and explain how initialization sensitivity is managed in the clustering process?


**Misc**:
- **Missing Error Metrics**: Figure 3 lacks error colormap ranges, making it difficult to assess error variability.
- **Physics-Inspired Segmentation**: Consider exploring modal analysis in engineering and fracture modes in graphics for relevant physics-driven segmentation methods.


**Rebuttal**:

First off, I deeply apologize for missing the deadline to reply as I’ve been travelling. I have reviewed the revised manuscript and sincerely appreciate the additional statistics provided by the authors. I also agree with Reviewer YvcY that the statistical analysis convincingly demonstrates that segmentation is indeed highly useful. Kudos to the authors for addressing the issues raised in such a short amount of time.

I believe my earlier discussion with the AC is relevant here, so I will share part of it for context.

---

Here, I want to raise my concerns. My expertise lies primarily in FEM for mesh-based methods and neural field/PINN approaches for physics + ML, which informed my review from the perspective of norms in mechanical engineering and computer graphics.

While the authors provided additional statistics in their rebuttal, which I found convincing, they failed to adequately address or reinforce the concerns I raised:

1. **Literature review**: The paper does not sufficiently engage with physics-inspired methods, particularly modal analysis, which has a long history in computational physics and mechanical engineering and has seen renewed interest in graphics (e.g., [[Benchekroun et al. 2023]](https://www.dgp.toronto.edu/projects/fast_complementary_dynamics_site/) and [[Sellan et al. 2022]](https://www.dgp.toronto.edu/projects/breaking-good/), and I am not suggesting to cite those papers, just want to highlight what **"physics-informed features from geometry ONLY"** *should* be claimed). Using a graph-based clustering algorithm for a physics problem with no physical grounding feels underwhelming and disconnected from established practices.

2. **Clustering methodology**: The rebuttal essentially agrees that there is no physically motivated or elegant method to determine the optimal number of clusters for initial clustering [(reference)](https://openreview.net/forum?id=pzasy8KRWK&noteId=FGgmSZdP0U). This makes it hard for me to imagine this method being applied to any serious physics/engineering problems.

Although the paper incorporates some good physics intuition, I struggle to view it as motivated by a desire to develop a true physics solver. Instead, it appears to explore clustering on graphs—a valid direction for follow-up work, but not convincingly framed here as a physics-based contribution.

---
I also found the exchange between the authors and Reviewer YvcY regarding whether the segmentation is "physically-informed" highly interesting [(the exchange can be found here).](https://openreview.net/forum?id=pzasy8KRWK&noteId=UlcjK7vO4H). However, while the geometric segmentation may have been effective for a *specific* simulation setup, this is a *correlation*, not *causation*. Furthermore, I question whether the method would generalize to the same geometric setup under different boundary conditions or initial conditions (again, sorry for the late reply, it is no longer possible for authors to provide more updates on this).

To avoid confusion, I strongly recommend refraining from describing the segmentation approach as "physical." While this may seem like a matter of semantics, precise language is essential in technical writing to ensure clarity and avoid misrepresentation.

Given these considerations, I will not change my rating for the paper. That said, I greatly appreciate the professional conduct of the authors and the significant improvements made to the paper during the rebuttal process.

---

> ### Author Response · Authors · 2024-11-22
> **Response to Reviewer 5KUk (part 1)**
>
> We sincerely thank the reviewer for their time, valuable feedback, and thoughtful suggestions. Below, we provide detailed responses to the questions and comments raised:
> > Q1: **Mesh Continuity**
> - We thank the reviewer for valuable feedback regarding the inclusion of additional metrics for mesh evaluation. Following the recommendation, we have incorporated Aspect Ratio and Chamfer Distance as additional metrics in the paper (details in Appendix D.1). These metrics provide a broader and more standardized assessment of mesh quality and geometric fidelity. We have updated all relevant tables (Table 3 to Table 7) to include these two metrics for completeness. Additionally, these metrics are utilized to guide the evaluation of the various ablation studies added, as detailed in Appendix E.2.
>
> > Q2: **Underwhelming PE Impact**: Could the authors elaborate on the observed impact of PE?
>
> We’ve done thorough analysis on the effect of PE and the details can be found in Appendix E.2. Below are our observed findings regarding PE:
> - [*Effectiveness of PE Relates to Segment Counts*]: In both the CylinderFlow and DeformingPlate datasets, incorporating PE with fewer segments enhances performance across multiple metrics by reducing positional ambiguity. When segment counts are low, each segment encompasses larger and more diverse areas, limiting the model's spatial detail and understanding of segment relationships. PE provides explicit positional information, enabling the model to differentiate between distinct regions within the same segment and better comprehend how segments neighbor and interact with each other. In contrast, as the number of segments increases, the inherent spatial resolution improves, and the benefits of PE decrease. In some cases, PE may introduce unnecessary complexity that hinders performance.
> - [*Dataset-Specific Factors*]: The Deforming Beam dataset, with its complex geometry and deformation, did not benefit from PE. This suggests that the effectiveness of PE is not only dependent on segment count but also on how well the PE implementation aligns with the dataset's specific characteristics.
> - [*Need for Tailored PE Approaches*]: For complex systems, a generic PE may not suffice. Customized positional encoding strategies that consider the unique geometry and deformation patterns might be necessary to realize large performance gains.
> - [*Conclusion*]: While PE unquestionably enhances the performance of graph-based networks, supported by strong theoretical foundations and an increasing volume of research, further mathematical advancements are crucial. Identifying optimal encoding strategies is essential for achieving consistent improvements, particularly when applying PE to mesh segments in a wide range of dynamic systems.

---

> > ### Author Response · Authors · 2024-11-22
> > **Response to Reviewer 5KUk (part 2)**
> >
> > > Q3: **Selection of Optimal Number of Clusters:** Could the authors expand on their criteria for selecting the optimal number of clusters, and explain how initialization sensitivity is managed in the clustering process?
> >
> > **Response regarding criteria for selecting the optimal number of clusters:**
> > - We appreciate the opportunity to specifically illustrate the process of selection optimal number of cluster/segment. Besides Table 4, we also analyzed the effect of number of segment to different metrics in Figure 7. To have a more comprehensive evaluation, we make a line plot (shown in Figure 9(d)) and consider several additional metrics (e.g. Chamfer Distance, etc) to illustrate the relationship between segment number and overall performance. Below we shown in details regarding how optimal cluster count can be determined.
> > - Based on the analysis of the evaluation metrics for the Deforming Beam dataset, 19 segments stand out as the optimal number. At this segmentation level, the model achieves the lowest RMSE and Chamfer Distance, indicating high prediction accuracy and precise shape representation. The Hausdorff Distance is also minimized, reflecting excellent alignment between the predicted and true meshes. While the Silhouette score peaks at 9 segments—suggesting well-defined and compact clusters—the slight decrease at 19 segments is offset by significant gains in other performance metrics. Choosing a lower number of segments, such as 3 or 9, may result in higher Silhouette scores but can compromise mesh detail and prediction accuracy due to insufficient spatial granularity. Conversely, selecting a higher number of segments beyond 19 shows diminishing returns, with only marginal improvements or slight degradations in some metrics and a continued decline in Silhouette scores, potentially indicating over-segmentation and unnecessary computational complexity.
> > - When presented with a new dataset, the optimal number of segments can be determined by first computing segmentation quality metrics, such as Silhouette scores, for various segment counts to assess cluster cohesion and separation without requiring model training. This provides initial guidance on meaningful segmentation levels. Subsequently, training the model with varying segment numbers and evaluating performance metrics like RMSE, Hausdorff Distance, and Chamfer Distance will help identify the point where performance improvements plateau or begin to reverse, indicating the optimal balance between segmentation detail and model efficacy.
> >
> > **Response regarding how initialization sensitivity is managed in the clustering process:**
> > - Initialization sensitivity in our Physics-Guided Mesh Segmentation algorithm is effectively managed through a deterministic initialization process combined with iterative refinement. Specifically, we utilize METIS to generate the initial segmentation. METIS employs a multilevel graph partitioning approach that systematically coarsens the graph, partitions the coarsened graph, and then refines the partitions as the graph is progressively uncoarsened. This process relies on deterministic rules, such as heavy-edge matching and specific refinement strategies, ensuring consistent results across multiple runs with identical inputs. Given the same mesh and configuration, METIS produces consistent and reproducible segments. This determinism eliminates variability caused by random initializations, providing a stable and reliable starting point for the segmentation process. Following initialization, our algorithm employs an iterative refinement phase that leverages physics-guided features to dynamically adjust and optimize the segments based on the system's dynamic behavior. These combined approaches ensure stable and accurate segmentations, enhancing the algorithm’s robustness and performance across diverse mesh configurations and dynamic systems.
> >
> > > Q4: **Missing Error Metrics:** Figure 3 lacks error colormap ranges, making it difficult to assess error variability.
> > - We thank the reviewer for bringing this to our attention and we've added colormap ranges to the figure to better assessment.

---

> > > ### Author Response · Authors · 2024-11-22
> > > **Response to Reviewer 5KUk (part 3)**
> > >
> > > > Q5: **Physics-Inspired Segmentation:** Consider exploring modal analysis in engineering and fracture modes in graphics for relevant physics-driven segmentation methods.
> > > - We thank the reviewer for the valuable feedback and suggestions. We acknowledge that the current segmentation approach may be highly dependent on the initial setup, potentially missing important physical interdependencies. While our method currently employs static segmentation, it is feasible to implement dynamic re-segmentation, either after each time step or periodically, to refine mesh segments based on evolving dynamics. This approach could be particularly beneficial in scenarios involving rapid or significant deformations, where the model could detect such changes and trigger re-segmentation as needed, while retaining the initial segmentation when appropriate.
> > > - Additionally, we appreciate the suggestion to explore modal analysis in engineering and fracture modes in graphics for physics-driven segmentation methods, and we will definitely investigate these aspects in our future work.

---

> > > > ### Author Response · Authors · 2024-12-02
> > > > **Message to Reviewer 5KUk**
> > > >
> > > > Dear Reviewer 5KUk:
> > > >
> > > > We would like to thank you again for your valuable feedback on our paper. We've revised the manuscript, adding new experiments and analyses to strengthen it based on your suggestions. We kindly invite you to review these updates to see if they address your concerns. As the discussion period will be ending soon, we sincerely hope to have the opportunity to receive any additional suggestions you may have and address any additional questions or concerns.
> > > >
> > > > Best regards,
> > > >
> > > > Authors of Paper 3197

---

### Official Review · Reviewer_wGac · 2024-11-04

**Soundness:** 3
**Presentation:** 3
**Contribution:** 3
**Rating:** 6
**Confidence:** 4

**Summary:**

The paper presents the Mesh-based Multi-Segment Graph Network (MMSGN), a hierarchical model for simulating dynamic systems by capturing both local and global interactions in mesh-based structures. It addresses the challenge of balancing accuracy and computational efficiency, which is often an issue in existing Graph Neural Network (GNN) models used for dynamic simulations. The MMSGN framework utilizes a physics-guided hierarchical information exchange, merging micro-level node interactions and macro-level segment exchanges aligned with physical properties through a transformer. This approach allows for scalable, accurate modeling of complex behaviors and long-range dependencies, validated across several datasets, including a new DeformingBeam dataset designed to test long-range interactions and scalability.

**Strengths:**

+ The dual-level approach combining micro-level and macro-level interactions enables MMSGN to accurately model both short- and long-range effects, which improves accuracy across datasets.
+ The method can be used for both Lagrangian and Eulerian systems.
+ Segmenting the mesh based on physical properties through a transformer ensures that nodes within each segment exhibit similar behaviors, leading to better model convergence and reduced boundary discontinuities.
+ The model generalizes well to larger and denser mesh configurations, as demonstrated on the DeformingBeam large-scale dataset.
+ MMSGN preserves mesh fidelity and continuity better than competing models

**Weaknesses:**

- The model does not enforce strict physical constraints at contact points, potentially causing overlapping meshes in certain configurations.
- Achieving optimal performance requires careful tuning of segmentation and message-passing parameters, which may limit ease of application in new scenarios.
- The segmentation-based approach may lead to minor inconsistencies at boundaries, especially in cases involving diverse material properties.

**Questions:**

1. The paper does not discuss run time for the different configurations. It would be good to include some measure of the computational performance.

---

> ### Author Response · Authors · 2024-11-22
> **Response to Reviewer wGac**
>
> We sincerely thank the reviewer for their constructive feedback, which offers valuable guidance for future research. We also appreciate their recognition of the strengths of our paper. We recognize that adding physical constraints at contact points and enforcing boundary consistency, especially for systems with diverse material properties, are important areas for improvement. These enhancements are part of our planned future work to further refine our method. We believe addressing these points will enhance the robustness, flexibility, and applicability of our method in more complex scenarios.
>
> > Q1: Achieving optimal performance requires careful tuning of segmentation and message-passing parameters, which may limit ease of application in new scenarios.
> - Regarding hyperparameter selection, we acknowledge the challenge it poses and have added a section to the paper providing a guideline for selecting optimal segment numbers (please refer to Appendix E.2). This guideline aims to assist users in achieving better performance with reduced trial-and-error efforts.
>
> > Q2: The paper does not discuss run time for the different configurations. It would be good to include some measure of the computational performance.
> - The run times for training and testing across different models and datasets are reported in Table 8. Additionally, Section 4.2.2 includes a discussion of computational efficiency, where we compare different models in terms of accuracy and efficiency. If more detailed information is needed, such as run times for specific model hyperparameters or architectures, we would be glad to provide additional analysis or supplementary materials.

---

> > ### Author Response · Authors · 2024-12-02
> > **Message to Reviewer wGac**
> >
> > Dear Reviewer wGac:
> >
> > Thank you once again for your valuable feedback on our paper. In addition to addressing your questions, we have revised the manuscript by incorporating new experiments and analyses to strengthen it. We kindly invite you to review these updates to see if they help address some of your concerns from a different perspective. As the discussion period is drawing to a close, we sincerely hope to receive any additional suggestions you may have and to address any remaining questions or concerns.
> >
> > Best regards,
> >
> > Authors of Paper 3197

---

### Official Review · Reviewer_Kw4Q · 2024-11-04

**Soundness:** 3
**Presentation:** 3
**Contribution:** 3
**Rating:** 6
**Confidence:** 3

**Summary:**

In this paper, the authors present a Mesh-based Multi-Segment Graph Network (MMSGN) aiming to address the challenges in learning dynamic system, such as achieving high accuracy and efficiency in scenarios with complex mesh structures, and extensive long-range effects. The proposed method integrates micro-level local interactions with macro-level global exchanges, aligning the hierarchical mesh structure to reflect the system’s physical properties, enabling it to effectively capture both local and global dynamics. The method is validated  on several dynamic datasets and benchmarks with existing methods. The experiments show that the MMSGN demonstrates advantages on accuracy, mesh quality and managing long-range effects on test cases. The MMSGN also shows generalization ability, which can be applied to larger physical systems.

**Strengths:**

The paper is well written and organized. The idea of exchanging between micro-level and macro-level is intuitive. Inspiring by domain decomposition in physical simulation, the physics-guided mesh segmentation use not only the spatial distance but also the similarity between features for mesh nodes clustering, which looks sound. Experiments are solid and there are extensive comparisons to benchmarks. The proposed method is demonstrated to be effective on metrics including prediction error and mesh quality comparing to the benchmarks.

**Weaknesses:**

I didn't see obvious weaknesses for this paper. The authors claim that the proposed method can be applied on large cases. However in Table 2 in appendix, the largest scenario DeformationBeam (large) contains 4540 nodes, which is relatively not large comparing to real world cases (usually more than 10k or even 1M). It is unclear how the proposed method perform on these large scale scenarios.

**Questions:**

- In Mesh Segment Feature Dispatch part, how does the extracted micro-level information from MeshGraphNet integrates  with the macro-level features extracted from the mesh segments ?
- In Line 219 to Line 220, it mentions that "prior information I" includes boundary conditions, material properties etc. How the "prior information I" mentioned in line 249 calculated? How does these information utilized in the Physics-guided Segmentation part? It seems that there are only distance information(distance to obstacle nodes and distance to boundary nodes) included as described.
- In Line 348, it mentions that "predicted mesh in Lagrangian systems", just wondering what the predicted mesh indicates? Does it mean the method also output a mesh in addition to the current states on each nodes of a mesh?
- Does the segmentation performed every simulation step or just performed once? If is is only performed once, does the mesh segmentation still effective during the simulation especially for problems potential with large mesh deformation?
- How does the number of segments influence the performance of the proposed method?
- How does the boundary conditions imposed in this pipeline?
- How long is the training time and how much memories required for training of this method?
- How will the method perform on a case which is more similar to a real world case (such as 1M)?

---

> ### Author Response · Authors · 2024-11-22
> **Response to Reviewer Kw4Q (part 1)**
>
> We sincerely thank the reviewer for taking the time to review our paper and provide valuable feedback. We also appreciate the reviewer’s recognition of our work. Below, we address the questions and comments raised in detail:
> >Q1: In Mesh Segment Feature Dispatch part, how does the extracted micro-level information from MeshGraphNet integrates with the macro-level features extracted from the mesh segments?
> - For mesh node $i$, the integration is performed by concatenating its micro-level feature $\mathbf{h}_i$, obtained from the  Encoder-Process-Decode (EPD) structure, with the macro-level feature  ${{\mathbf{h}}_S}_i$, where $S_i$ is the segment that $i$ is assigned to. We found this straightforward integration efficient and effective for dynamic system prediction.
>
> > Q2: How the "prior information I" mentioned in line 249 calculated? How does these information utilized in the Physics-guided Segmentation part?
> - In our work, the prior information $I$ for the three datasets includes boundary information (e.g., walls in the Cylinder Flow dataset, edges of deformable materials) and obstacle information. These are important because they define critical physical boundaries and obstacles that significantly influence the system's dynamics, affecting how forces and flows propagate through the mesh.  We incorporate this prior information by defining functions to calculate prior-related features, as described in Eq (4) of our paper. Specifically, we use various distance measures—such as distances to obstacle nodes and boundary nodes—to compute these features, which are then utilized by our Physics-guided Segmentation module to guide the segmentation process.
> - While we employ distance measures in this study, other features can also be used to represent prior information, such as material property variations, stress or strain distributions, or external force fields. These alternatives can provide additional insights into the physical behavior of the system and further inform the segmentation. In our datasets, we focused on single-material deformable objects, so material-related prior information was not included. However, our method can easily incorporate material-related features when simulating multi-material systems with different properties and behaviors. In such cases, the segmentation method would adjust based on material properties—for example, segmenting hyperelastic materials by regions of high elastic deformation, and plastic materials by areas undergoing permanent deformation due to yielding.
> - In conclusion, our method is versatile and capable of incorporating various priors and user-defined features to guide segment selection. We plan to explore the integration of additional priors, such as material properties and stress distributions, in future work to enhance the segmentation's physical relevance.
>
> > Q3: It mentions that "predicted mesh in Lagrangian systems", just wondering what the predicted mesh indicates? Does it mean the method also output a mesh in addition to the current states on each nodes of a mesh?
> - By "predicted mesh," we refered to the predicted states of the mesh nodes, rather than the generation of a separate or additional mesh. In Lagrangian systems, the positions of mesh nodes evolve over time. To evaluate the quality of these predictions, we introduced metrics that assess the mesh quality based on the predicted node positions. To improve the clarity, we have modified "predicted mesh" to "mesh with the predicted node positions" in our manuscript.

---

> > ### Author Response · Authors · 2024-11-22
> > **Response to Reviewer Kw4Q (part 2)**
> >
> > >Q4: Does the segmentation performed every simulation step or just performed once? If is is only performed once, does the mesh segmentation still effective during the simulation especially for problems potential with large mesh deformation?
> > - In our approach, segmentation is performed once at the initial time step, and the simulation proceeds using this initial segmentation, reflecting a realistic scenario. To demonstrate its effectiveness during large deformations, we have included additional visualization results in Figure 6, where mesh nodes at each predicted time step are colored based on the average prediction error within their segments. These results show that our method produces segments with uniform colors across time steps, indicating that nodes within the same segment share similar dynamic behaviors and different segments have little discrepancies or maintain high continuity. This means that our segmentation effectively groups regions with coherent dynamic interactions, ensuring consistent modeling and accurate prediction of the system's evolution over time.
> > - While our current method uses static segmentation, it is indeed possible to perform re-segmentation after every time step or periodically to refine the mesh segments based on the latest dynamics. Dynamic re-segmentation could be more effective in cases where large deformations occur rapidly. The model could detect such behavior and trigger re-segmentation as needed, otherwise retaining the initial segmentation.
> >
> > > Q5: How does the number of segments influence the performance of the proposed method?
> > - Table 5 and Table 6 present the RMSE-1, RMSE-all, and various mesh quality metrics as the total number of mesh segments is varied during training on three different datasets. In general, MMSGN maintains stable performance with relatively low variance, indicating that results are not highly sensitive to segment count. This robustness ensures reliable accuracy across different mesh granularities. However, increasing the number of segments—thereby reducing finite elements per segment—can lead to slight decreases in accuracy and performance.
> > - To comprehensively evaluate the effect of segment number and determine the optimal segmentation for a given dataset, we analyzed prediction accuracy across a wide range of segment counts (from 3 to 51) on the DeformingBeam dataset. The impact of varying the number of mesh segments on prediction accuracy is illustrated in Figure 8 and Figure 9(d). Please see our updated results and detailed discussion regarding the effect to segment numbers in Appendix E.2 Influence of Segment Count on Performance.
> >
> > > Q6: How does the boundary conditions imposed in this pipeline?
> > - Boundary conditions are incorporated by encoding the node type into the input features, allowing the model to differentiate boundary mesh nodes from others and account for their distinct influence on the dynamic system. Among the datasets investigated, CylinderFlow adopts fixed values (e.g. zero velocity) for boundary nodes; DeformingPlate and DeformingBeam maintain fixed positions for boundary handle nodes. During rollout, MMSGN explicitly enforces these boundary conditions by assigning the fixed values to the relevant boundary nodes, ensuring consistency with the physical setup.
> >
> > > Q7: How long is the training time and how much memories required for training of this method?
> > - Please find the training time and memories summarized in Table 8 for our method and other state-of-the-art methods.

---

> > > ### Author Response · Authors · 2024-11-22
> > > **Response to Reviewer Kw4Q (part 2)**
> > >
> > > > Q8: How will the method perform on a case which is more similar to a real world case (such as 1M)?
> > > - To the best of our knowledge, there are currently no publicly available simulation datasets of this scale, likely due to the significant computational resources and time required to generate and process such extensive data. However, we recognize the importance of addressing the scalability of our method to real-world applications.
> > > - For such large systems, computational limitations, particularly in memory usage, become a critical concern. To handle this, our approach can be adapted to partition the large mesh into smaller subdomains or patches. Each subdomain would contain a reduced number of nodes, comparable to or slightly larger than the datasets we have already experimented with. Within each subdomain, our method can be applied as demonstrated, effectively modeling the local dynamics. To ensure seamless continuity and compliance with contact and boundary conditions across subdomains, we can employ coupling techniques such as domain decomposition methods or overlapping Schwarz methods. These techniques facilitate the interaction between adjacent subdomains, allowing for the accurate simulation of the global system behavior without the need to process the entire mesh simultaneously.
> > > - Our current work serves as a foundational step toward this goal by demonstrating the effectiveness of our method—particularly in computational efficiency, prediction accuracy, and generalizability—on smaller-scale systems. By validating our approach in these contexts, we have established its potential for scalability. Extending our method to larger domains by incorporating advanced partitioning strategies and inter-subdomain communication protocols would be a compelling area for future exploration.

---

> > > > ### Comment · Reviewer_Kw4Q · 2024-11-25
> > > > **response**
> > > >
> > > > Thanks for the authors' detailed reply. I would keep my score unchanged.

---

> > > > > ### Author Response · Authors · 2024-11-25
> > > > >
> > > > > We appreciate the reviewer taking the time to evaluate our detailed responses and for recognizing the value of our work. Should the reviewer have any further questions or require additional clarifications before the discussion period ends, we would be more than happy to assist.

---

### Meta-Review · Area_Chair_t4Zz · 2024-12-20

**Metareview:**

The submitted paper proposed a hierarchical GNN-based solution for physics problems on meshes based on the difference between micro-level exchanges and macro-level exchanges, with the separation between these two obtained by a data-driven clustering process, whose features are informed by geometry (scene structure). It received five reviews, and stayed borderline during the peer-review phase. No clear consensus was reached at the end.

The reviewers appreciated the good performance of the method, with sound comparisons, and the idea of the dual level approach (more on this claim further below), generalization, and the fact that the paper was well-written and easy to read.

The main weaknesses raised were

- Novelty (the method is very close to an existing method of the literature), clarity of the contributions,
- Misleading claims, and overstatement on the physics-informed nature of the clustering process, which in reality only uses geometric features extracted from the ground-truth scene geometry,
- misleading or wrong terminology,
- limitations of a segmentation based method, eg. in presence of moving objects,
- doubts on the generalization of the segmentation method over datasets and hyper parameters,
- requirements on tuning the method,
- sensitivity to mesh and/or segment resolution,
- missing analysis,

The authors provided extensive answers which satisfied some but not all reviewers.

In the next phase, the discussion between reviewers and including the AC was then dominated by two aspects: the overclaims of physics-informed clustering/segmentation, and the contribution of micro-level exchanges vs. macro-level exchanges. Reviewers YvcY and 5KUk were most critical of the work, with reviewer 5KUk considering lowering the score to 3 (from 5). The AC sides with the critical reviewers and judges that the paper is not yet ready for publication. Key to this  assessment were the following points:

- The main arguments for acceptance mentioned by the positive reviewers where the separation between micro-level and macro-level exchanges, which was the main contribution identified by the positive reviewers. However, these claims have been refuted multi times by the critical reviewers and the AC, as this type of separation is an established methodology (Liu et al, 2021, Janny et al 2023, Cao et al, 2023). The positive reviewers seem to have been unfamiliar with this literature, and repeated questions by the AC to confirm positioning were ignored by them.
- A large-part of the paper's orientation and positioning is the physics-informed nature of the clustering module, but this claim does not hold. While the AC agrees with reviewer BRQ5 that the physical nature is not in a requirement for the method to work (the hierarchical methods of the literature are also not physics-informed), it indeed is a central claim of the paper and is even part of the paper's title. As such, the claims of the paper are an overstatement and potentially misleading.

It is the AC's assessment, that the positive aspects of the paper (a slight change of the clustering process compared to the closest method, as well as change in positional encoding, leading to better performance) does not compensate for the flaws of the paper, which are quite significant overstatements in positioning, contributions, and origins of the gains.

**Additional Comments On Reviewer Discussion:**

The reviewers engaged with the authors, and discussed the paper with the AC.

---

### Decision · Program_Chairs · 2025-01-22

Reject